# Crossing the blood-brain-barrier with nanoligand drug carriers self-assembled from a phage display peptide

Lin-Ping Wu [1]*, Davoud Ahmadvand[2,8], Junan Su[3], Arnaldur Hall[4], Xiaolong Tan[5], Z. Shadi Farhangrazi[6] & S. Moein Moghimi [2,5,7,9]*

The filamentous bacteriophage fd bind a cell target with exquisite specificity through its few copies of display peptides, whereas nanoparticles functionalized with hundreds to thousands of synthetically generated phage display peptides exhibit variable and often-weak target binding. We hypothesise that some phage peptides in a hierarchical structure rather than in monomeric form recognise and bind their target. Here we show hierarchial forms of a brain-specific phage-derived peptide (herein as NanoLigand Carriers, NLCs) target cerebral endothelial cells through transferrin receptor and the receptor for advanced glycation-end products, cross the blood-brain-barrier and reach neurons and microglial cells. Through intravenous delivery of NLC-β-secretase 1 (BACE1) siRNA complexes we show effective BACE1 down-regulation in the brain without toxicity and inflammation. Therefore, NLCs act as safe multifunctional nanocarriers, overcome efficacy and specificity limitations in active targeting with nanoparticles bearing phage display peptides or cell-penetrating peptides and expand the receptor repertoire of the display peptide.

[1] Guangzhou Institute of Biomedicine and Health, Chinese Academy of Sciences, Guangzhou 510530, People's Republic of China. [2] Nanomedicine Laboratory, Centre for Pharmaceutical Nanotechnology and Nanotoxicology, Department of Pharmacy, Faculty of Health and Medical Sciences, University of Copenhagen, Universitetsparken 2, DK-2100 Copenhagen Ø, Denmark. [3] Hefei Institute of Stem Cell and Regenerative Medicine, Guangzhou Institute of Biomedicine and Health, Chinese Academy of Sciences, Guangzhou 510530, People's Republic of China. [4] Genome Integrity Unit, Danish Cancer Society Research Center, Copenhagen, Denmark. [5] School of Pharmacy, King George VI Building, Newcastle University, Newcastle upon Tyne NE1 7RU, UK. [6] S. M. Discovery Group Inc., Denver, CO, USA and S. M. Discovery Group Ltd., Durham, UK. [7] Institute of Cellular Medicine, Faculty of Health and Medical Sciences, Framlington Place, Newcastle University, Newcastle upon Tyne NE2 4HH, UK. [8] Present address: School of Allied Medical Sciences, Iran University of Medical Sciences, Tehran, Iran. [9] Present address: School of Pharmacy, King George VI Building, Newcastle University, Newcastle upon Tyne NE1 7RU, UK, and Institute of Cellular Medicine, Framlington Place, Newcastle University, Newcastle upon Tyne NE2 4HH, UK. *email: linping.wu@hotmail.com; seyed.moghimi@ncl.ac.uk

Nanoparticulate drug carriers are receiving considerable attention for site-specific targeting of diagnostic and therapeutic agents for detection, monitoring and treatment of various diseases[1]. A dominant strategy in targeting with drug carriers is their surface functionalisation with a wide range of target-specific ligands (e.g., folate, antibodies and their fragments, proteins such as transferrin and a plethora of designer peptides), a process often referred to as "active targeting"[2–4].

The phage display technology is a valuable tool in identification of numerous peptide ligands for active targeting[5–10]. Here the library of foreign peptide or protein variants is displayed as expression of the fusions to the bacteriophage coat protein (e.g., the p3 protein on the bacteriophage fd) and each variant is shown on the surface of a virion[5]. On biopanning and recovery of the strongly bound phage particles, each peptide ligand is identified by DNA sequencing and then synthesised[5]. Efforts in active targeting with drug carriers decorated with synthetically produced phage-derived peptides, however, is met with varying levels of success[3,4]. One example is the filamentous phage fd clones displaying the 15-amino acid peptide GYRPVHNIRGHWAPG (herein as GYR peptide) on its minor coat protein (p3), which shows high binding specificity to human brain capillary endothelial hCMEC/D3 cells and the mouse brain endothelium in vivo[8]. On the other hand, liposomes bearing the linear form of GYR show weak target engagement regardless of the peptide density on liposome surfaces[11]. Numerous factors such as particle shape, possible changes to peptide structure and orientation during functionalization as well as ligand masking by plasma/serum proteins may explain these discrepancies[3,4,11–14]. In addition to these, a different peptide arrangement in the phage microenvironment[5,15,16] may account for these discrepancies and simply grafting a synthetic phage peptide to drug carriers may not necessarily simulate those architectural display at the bacteriophage tip that confer target recognition and specificity. Indeed, the filamentous phage fd has five copies of the p3 protein on its surface and in close proximity to each other[5]. Interdomain interactions and disulfide bridges within the p3 protein can influence p3 organisation, which in turn may promote intermolecular bonding among adjacent display peptides to form a multivalent domain with high target binding avidity[15–17]. This hypothesis might be applicable to the GYR peptide, where adjacent GYR peptides on p3 scaffolds could potentially interact with each other through arginine (R)-mediated bidentate hydrogen bonding, non-covalent π stacking between neighbouring aromatic indole side-chain of tryptophan (W) and imidazole of histidine (H) and salt-bridges. Thus, supramolecular forms of GYR, rather than in monomeric state, may confer improve target recognition. We test this hypothesis by directing GYR self-assembly into core-shell nanoparticles and multiple crossed β-sheet nanofibrils [thereafter as NanoLigand Carriers (NLCs)]. NLCs target at least two receptors on cerebral endothelial cells, cross the blood-brain-barrier (BBB) on intravenous injection and deliver functional nucleic acids into the brain without adverse reactions and toxicity. Thus, hierarchical forms of a brain-specific phage peptide overcome specificity limitations in active targeting and crossing of the BBB with conventional nanoparticles[11,18,19].

## Results

**Peptide synthesis and self-assembly.** We performed a two-step chemical modification at the *N*-terminal of GYR to initiate self-assembly. First, we added a cysteine (C) amino acid to promote cross-linking (via disulfide bridge formation) between two adjacent peptides. Next, we linked a fluorophore (5-carboxy-fluorescein, 5-FAM, or cyanine 5.5, Cy5.5) to the amino group of C to initiate π–π stacking among conjugates. Fluorophore introduction also allows for qualitative monitoring and quantification of peptide uptake by cells and in vivo tracing. The fluorescent amphiphile-peptide conjugate, termed FAM-CGY (Table 1), was synthesised by a standard solid-phase method, purified by semipreparative reverse-phase high performance liquid chromatography and characterised by analytical HPLC and mass spectrometry (Supplementary Fig. 1). FAM-CGY exhibited a molecular mass of 2075.24 g mol$^{-1}$, a critical aggregation concentration (CAC) of 2.8 μM in Milli-Q water (Supplementary Fig. 2) and a β-sheet structure at concentrations above CAC (Supplementary Fig. 3). At concentrations below CAC, FAM-CGY (and CGY) displayed a random coil conformation (Supplementary Fig. 3). For comparison, we synthesised and characterised a number of FAM (or Cy5.5)-tagged analogues (Table 1, Supplementary Figs. 4–10). SDS-PAGE analysis revealed the presence of FAM-CGY monomer and dimer below the CAC, but oligomers were additionally present above the CAC (Fig. 1a). When tris(2-chloroethyl) phosphate (which reduces C to free thiol) was present, FAM-CGY did not form oligomers (Fig. 1b). Similarly, FAM-GYR (the analogue without C) did not form any detectable oligomers (Fig. 1c). On the other hand, in the absence of fluorophore, CGY only formed dimers (Fig. 1d). These observations suggest a role for disulfide bridges in dimerisation and fluorophore-mediated π–π stacking in oligomerisation processes. Re-positioning of FAM to the C-terminal, while maintaining the C at the N-terminal [CGY-(K)-FAM] or changing the position of arginine 10 (R10) in FAM-CGY (termed scrambled peptide 1, SP1), or replacing the single tryptophan 13 (W13) from the sequence with a glycine (G) (termed SP2) all prevented oligomerisation (Fig. 1e, f). These results further highlight the importance of sequence specificity in peptide self-assembly/oligomerisation, which may include R-mediated bidentate hydrogen bonding, histidine (H)-mediated hydrogen-π interactions, and additional π-π stacking contribution from the aromatic indole side-chain of W and imidazole in H. The *N*-terminal introduction of both FAM and C to a structurally irrelevant peptide of 8 amino

### Table 1 Peptide and fluorophore-peptide conjugate description

| Peptide Abbreviation | Amino acid sequence | Molecular mass (gmol$^{-1}$) | Purity (%) |
|---|---|---|---|
| GYR | GYRPVHNIRGHWAPG | Not applicable | – |
| CGY | CGYRPVHNIRGHWAPG | 1820.04 | 98.11 |
| FAM-CGY | 5-FAM-CGYRPVHNIRGHWAPG | 2178.36 | 98.91 |
| CGY-(K)-FAM | CGYRPVHNIRGHWAPGK-5-FAM | 2305.53 | 98.36 |
| FAM-GYR | 5-FAM-GYRPVHNIRGHWAPG | 2075.24 | 98.46 |
| FAM-CGY scrambled 1 | 5-FAM-CGYRPVHNIRGHWRAPG | 2177.36 | 98.56 |
| FAM-CGY scrambled 2 | 5-FAM-CGYRPVHNIRGHGAPG | 2048.20 | 98.26 |
| Cy5.5-CGY | Cy5.5-CGYRPVHNIRGHWAPG | 2385.30 | 98.20 |
| Cy5.5-CGY scrambled 1 | Cy5.5- CGYRPVHNIGHWRAPG | 2385.90 | 98.50 |

Sequences are represented by one standard amino acid code symbols. 5-FAM = 5-carboxy-fluorescein and Cy5.5 = cyanine 5.5

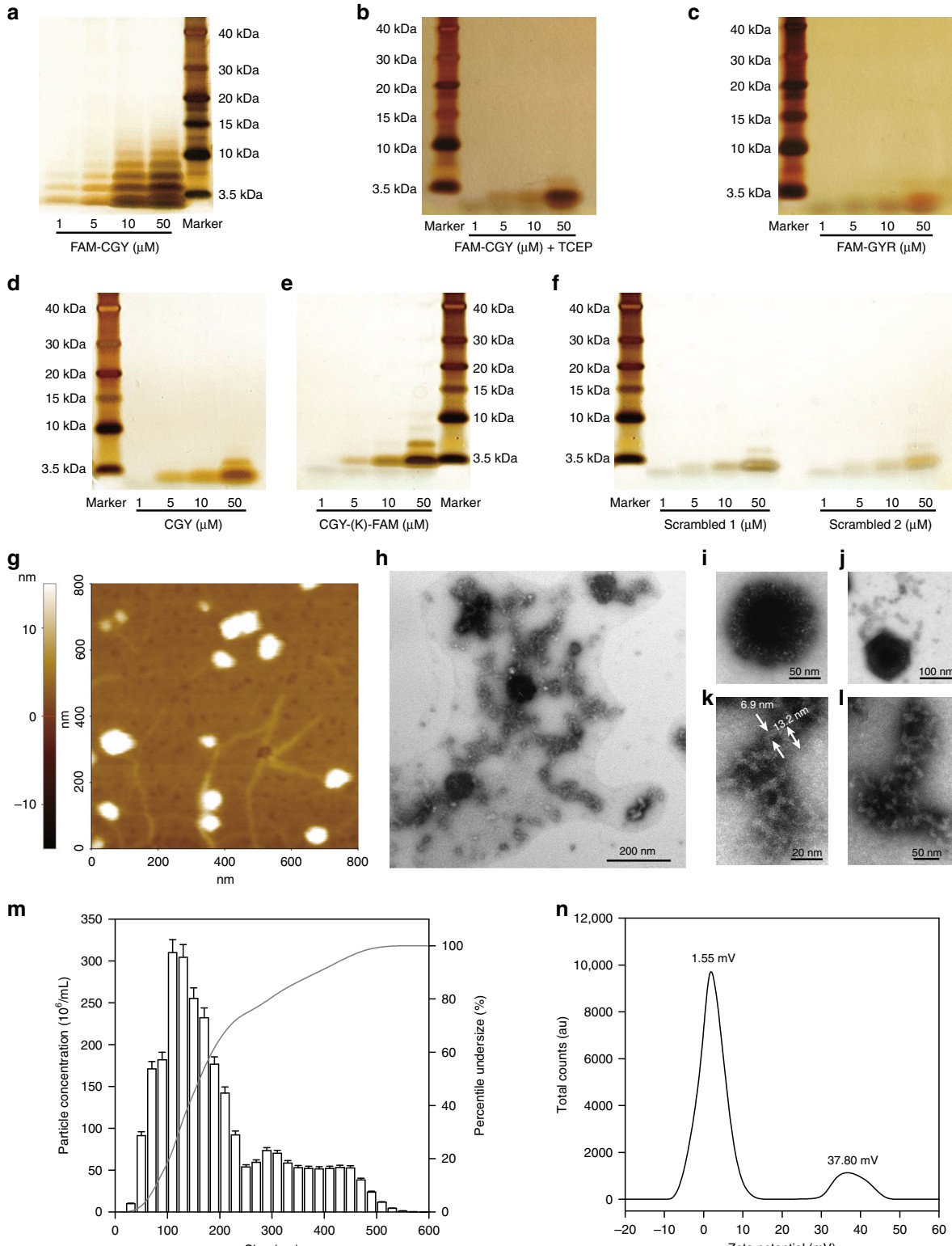

**Fig. 1** FAM-CGY assemblies and their characterisation. **a** Non-reducing SDS-PAGE of FAM-CGY at different concentrations showing oligomer formation at >5 μM. **b** SDS-PAGE of FAM-CGY in the presence of tris(2-chloroethyl) phosphate (TCP), which reduces cysteine to free thiol and prevents FAM-CGY dimer and oligomer formation. **c–f** SDS-PAGE analysis of FAM-GYR, CGY and other analogues, showing absence of oligomers. Peptide sequences are depicted in Table 1. Molecular weight markers are shown in each gel (last lane in a and e and first lane in other gels). **g** Representative topographic atomic force microscopy images of FAM-CGY assemblies. The image shows the presence of both nanoparticles and fibre-like structures (white threads). **h** Representative electron micrographs of FAM-CGY assemblies showing the presence of electron dense nanoparticles and branched-fibres. **i** Magnified electron micrograph of a typical nanoparticle, where tightly packed protofilament projections forms the shell. **j** Shows a nanoparticle with surface projected fibres. **k**, **l** Representative magnified views of twisted nanofibres. **m** A typical size distribution profile of FAM-CGY NLCs as a function of particle concentration determined by nanoparticle tracking analysis. **n** Calculated zeta potential values of FAM-CGY NLCs from electrophoretic mobility measurements. Each experiment was repeated three times with different preparations of FAM-CGY ($n = 3$) with identical results

acids (the NAP peptide that lacks R and H amino acids)[20], which is the smallest active element of activity-dependent neuroprotection protein that exhibits potent neuroprotection action, did not induce aggregation or formation of higher structures (Supplementary Fig. 11). Thus, we have demonstrated multifactorial processes in GYR dimerisation and oligomerisation.

**Characterisation**. Atomic force microscopy (AFM) and transmission electron microscopy (TEM) of FAM-CGY showed presence of both nanoparticles and fibre-like species (Fig. 1g, h). No structural arrangements were detected or observed with other analogues. The electron micrographs revealed a core-shell morphology for nanoparticles (Fig. 1i). The core component is presumably an assembly of highly condensed FAM-CGY multimers that grow further, eventually forming tightly packed protofilament projections of the shell component (Fig. 1i). Occasionally, protofilaments projections of the core-shell particles grow into fibrils (Fig. 1j) with twisted elongated architectures (Fig. 1k, l). These fibrils are presumably stabilised through extensive R-mediated bidentate hydrogen bonding (due to charge delocalisation on guanidium group in R) as well as FAM- and aromatic amino acid side-chain directed π–π stacking. These suggestions are in line with reported observations that have indicated a role for such interactive forces in forming fibres and sheets from synthetic peptide amphiphiles[21–24]. Thus, the inability of the scrambled peptides to form protofibrils and nanofibres further highlight the critical role of 5-FAM-peptide secondary structure (Supplementary Fig. 3) in multimer formation.

Energetically, β-sheet ribbons may be unstable, since the hydrophobic blocks of the fluorophore and some aromatic amino acid side-chains may be exposed to the solvent. However, twisting of β-sheet ribbons would bury such hydrophobic domains inside, resulting in stable fibril formation. Collectively, these arrangements may contribute to amphiphile-peptide self-assembly into cylindrical, ribbon-like, twisted single or multiple fibres (Fig. 1k, l)[21,25,26]. Morphologically, these elongated structures resemble many misfolded amyloid type protein aggregates in neurodegeneration, where two or more protofilaments twist repetitively around each other[27,28]. SDS-PAGE studies (Fig. 1a) also suggest FAM-CGY assemblies formed in Milli-Q water (including nanofibres) are not SDS-stable, since they appear as monomer through hexamer on the gels.

Next, we studied size distribution of the aforementioned NLCs formed in Milli-Q water by Nanoparticle Tracking Analysis (NTA). The results in Fig. 1m show a typical size distribution of tracked species displaying a mean size of $169 \pm 77$ nm, with a mode of 138 nm. The tailing portion of the size-distribution histogram may represent dismantled fibrils, since no spherical particles >200 nm was observed under the transmission electron microscope. In contrast to FAM-CGY, neither CGY nor FAM-GYR, nor scrambled peptides formed any detectable species by NTA (or TEM) even at concentrations as high as 20 μM. Finally, from electrophoretic mobility measurements we calculated two zeta potential (ζ) peaks for NLCs; a major peak of +1.55 mV and a minor peak of +37.8 mV, respectively (Fig. 1n). The higher ζ value, presumably represent architectural arrangements with multiple surface exposed R amino acid residues.

**NLC-cell interaction**. The results in Fig. 2a show concentration-dependent association of FAM-labelled NLCs with human brain capillary endothelial hCMEC/D3 cells. NLC association with endothelial cells was not affected by the presence of serum proteins (Supplementary Fig. 12), thus the binding specificity of NLCs is preserved in the presence of extracellular proteins. Studies with a panel of transport inhibitors demonstrated that the

NLC uptake is energy-dependent, multifaceted and involves clathrin- and caveolae-dependent endocytic processes as well as macropinocytosis (Fig. 2b)[29,30]. The involvement of different NLC internalisation pathways is consistent with the polarised nature of endothelial cells and may suggest involvement of different receptors in NLC recognition[30]. In contrast to FAM-CGY, the cellular uptake of FAM-GYR and other modified/scrambled FAM-conjugated peptides was relatively low (Supplementary Fig. 13). Collectively, the abovementioned observations suggest that NLC binding to hCMEC/D3 cells is specific and predominantly structure-based.

Live-cell fluorescent microscopy studies confirmed NLC internalisation (Fig. 2a) and localisation to early endosomes and lysosomal compartments at the peri-nuclear regions (Supplementary Fig. 14). However, considering a role for caveolin-mediated internalisation[30], we did not observe notable fluorescent overlay with other organelles such as the Golgi apparatus and endoplasmic reticulum (Supplementary Fig. 15).

The cerebral capillary endothelial cells, including the hCMEC/D3 cell line, widely express homodimeric transferrin receptors (TfRs)[30,31] and confirmed here (Supplementary Fig. 16). First, we investigate whether TfR plays a role in NLC recognition and uptake, since TfR-ligand complexes undergo clathrin-dependent endocytosis[30,31]. Competition studies in the presence of increasing concentration of a partially iron saturated human transferrin diminished NLC uptake by hCMEC/D3 cells at 37 °C (Fig. 2c). The highest tested concentration of transferrin (500 nM), however, inhibited NLC uptake by ~50% thus suggesting a role for TfR in NLC uptake. However, this partial inhibition could be either due to rapid TfR recycling, or higher avidity of NLCs for TfR and/or involvement of other receptors in NLC uptake. Next, we showed intracellular co-localisation of FAM-CGY NLCs with Texas Red-labelled transferrin (Supplementary Fig. 17). This provided further support for a clathrin-mediated uptake processe[30,31]. Since transferrin could reduce NLC uptake by hCMEC/D3 cells, we further studied NLC uptake following TfR down-regulation with a TfR-specific siRNA (Fig. 2d). Using a commercial transfectant, TfR expression in hCMEC/D3 cells was down-regulated by 70% without inducing cell death (the cell viability was >95% as determined by the Trypan Blue exclusion test). This partial TfR down regulation reduced both transferrin (positive control) and NLC uptake by hCMEC/D3 cells (Fig. 2d) and the NLC uptake results are in line with transferrin competition studies (Fig. 2c). Since, some NLC uptake still proceeds, downregulation of TfR may have not impaired endocytic efficacy of hCMEC/D3 cells. Finally, to further address the possible role of TfR in NLC uptake, we compared NLC uptake by two epithelial human cell lines, one with high expression of TfR (MCF-7, a human breast cancer cell line) and the other with poor expression of TfR (MCF-10A, a non-tumourigenic epithelial cell line of mammary gland/breast origin, which expresses only 10% of TfR compared with MCF-7 cells). The results in Fig. 2e show predominant NLC uptake by MCF-7 cells. To show that NLC uptake by MCF-7 cells is TfR specific, we performed a saturation study. The results in Fig. 2f show that saturation of TfR binding sites with holo transferrin at 4 °C dramatically diminishes NLC binding and clearly differentiates between NLC-specific binding to TfR and nonspecific interactions with plasma membrane. These observations, therefore, support the involvement of TfR in NLC recognition.

Next, we considered the signal-transduction pattern recognition receptor RAGE in NLC binding, since RAGE is widely expressed on cerebral endothelial cells, is localised in caveolae and binds a diverse repertoire of multiple crossed β-sheet fibril ligands including beta-amyloid (Aβ) oligomers[32–37]. It is therefore plausible that RAGE could bind to multiple crossed β-sheet

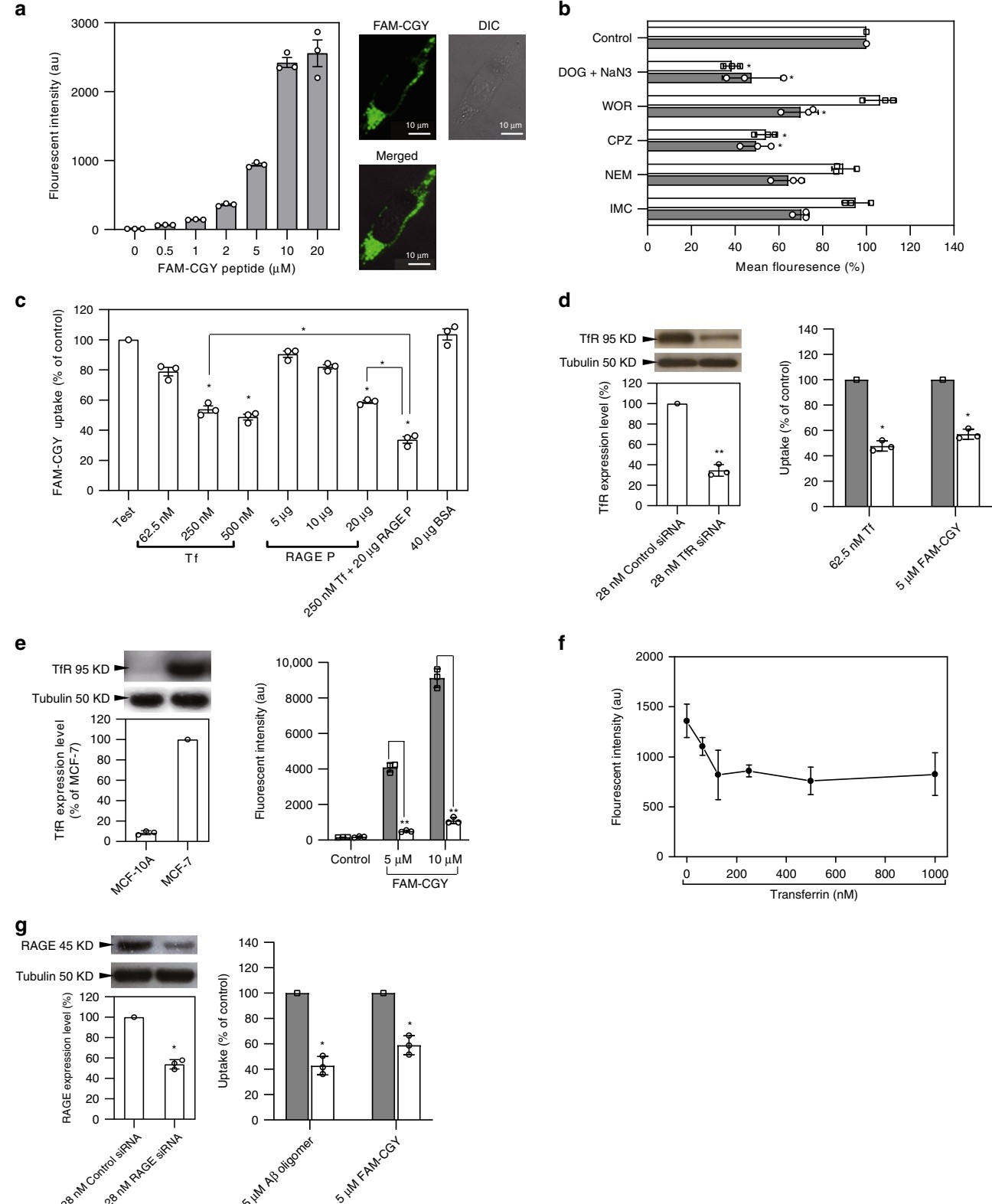

structures in NLC nanofibres and the protofilament components of the "core-shell" nanoparticles and direct internalisation. We tested this hypothesis by showing a significantly less ($p < 0.05$; non-paired two-sided student $t$-test) NLC uptake by hCMEC/D3 cells in competition with increasing concentration of a commercially available RAGE ligand (Fig. 2b). Also, lowering RAGE expression in hCMEC/D3 cells with a RAGE-specific siRNA (the

cell viability was >95% on RAGE down regulation as determined by the Trypan Blue exclusion test) reduced NLC uptake compared with their respective mock-silenced cells (Fig. 2g). Partial RAGE downregulation, also reduced tetramethylrhodamine (TAMRA)-labelled Aβ oligomer uptake by hCMEC/D3 cells, which is, at least, consistent with oligomeric Aβ$_{1-42}$ internalisation through endocytic processes and RAGE involvement in Aβ binding

**Fig. 2** NLC-cell interaction studies. **a** Concentration-dependent uptake of FAM-CGY assemblies by hCMEC/D3 cells at 24 h, 37 °C. The right panel shows a typical fluorescence microscopy and differential interference contrast (DIC) images of a live hCMEC/D3 cell post FAM-CGY (5 μM) challenge. **b** The effect of internalisation inhibitors on the uptake of FAM-CGY NLC (final concentration 5 μM) and Texas Red-labelled transferrin (5 μg per well) by hCMEC/D3 cells. The results represent mean fluorescent intensity with respect to Control (no metabolic inhibitor). Open column = transferrin; grey column = FAM-CGY. **c** FAM-CGY NLC (5 μM) uptake by hCMEC/D3 cells is competitively inhibited by partially iron saturated human transferrin (Tf) and a commercially propriety RAGE peptide (RAGE P). Test column = reference FAM-CGY NLC uptake (no inhibitors) with auto-fluorescence subtracted. Other incubations are compared with Test incubation. Bovine serum albumin (BSA) was used as an irrelevant protein. **d** Down-regulation of TfR in hCMEC/D3 cells with TfR-specific siRNA reduces Tf and FAM-CGY uptake. In the right panel, open column = control siRNA treated cells and grey column = TfR siRNA treated cells. **e** Comparison of TfR expression in MCF-7 and MCF-10A cells and FAM-CGY uptake. In the right panel, open column = MCF-7 cell line and grey column = MCF-10A cell line. **f** Diminished NLC (final concentration 5 μM) binding to MCF-7 cells at 4 °C on saturation of TfR binding sites. au = arbitrary unit. **g** The effect of RAGE downregulation in hCMEC/D3 with a RAGE-specific siRNA on TAMRA-labelled Aβ oligomer and FAM-CGY uptake. In the right panel, open column = control siRNA treated cells and grey column = RAGE siRNA treated cells. All incubations were done in triplicate and each experiment was repeated three times. Panels represent the mean value of three separate experiments ± s.d. and each dot indicates the mean of three technical replicates. *$p < 0.05$ and **$p < 0.01$, non-paired two-sided student $t$-test compared with respective controls. Source data are available in the Source Data file

(Fig. 2g). Considering RAGE involvement in Aβ uptake, we further showed competition between Aβ oligomers (globular structures of 5 nm in diameter)[37,38], and NLCs on cell uptake (Supplementary Fig. 18). However, we did not investigate as to whether this competition is directly through RAGE binding or could further involve other plasma membrane domains such as lipoprotein receptor protein and N-methyl-D-aspartate receptor, which are also known to bind Aβ[39]. Finally, simultaneous presence of both transferrin and the RAGE peptide ligand blocked NLC uptake more effectively (~70%) than any of the individual treatments in hCMEC/D3 cells (Fig. 2c), thereby confirming TfR and RAGE as predominant receptors in NLC uptake.

We therefore conclude that NLC uptake by brain endothelial cells involves at least two receptors (TfR and RAGE) and consistent with both clathrin- and caveolae-dependent modes of internalisation. Macropinocytosis, which has been implicated in the uptake of protein aggregates by different cells[40], may still account for the uptake of various oligomeric/aggregate forms of FAM-CGY. Our cell studies, however, could not differentiate on the mode of uptake between spheroidal and nanofibre NLCs at this stage.

Since hCMEC/D3 cell line has been validated as a BBB model[41], we sought to examine the effect NLCs on the integrity of this in vitro model. A hCMEC/D3 monolayer was formed and reached confluence on day 7 and the barrier integrity was confirmed through trans-endothelial electrical resistance (TEER) and cell layer capacitance (which reflects the membrane surface area) measurements with a CellZscope (Supplementary Fig. 19). TEER values were peaked on day 7 (~40 Ω cm$^{-2}$) in transwell inserts and longer culture of cells (up to 10 days) did not improve TEER and formation of multiple cell-layers was sometime observed from day 7 onward. Again, at day 7 the cell layer capacitance was below 2 μF cm$^{-2}$ (compared with ~10 μF cm$^{-2}$ in empty inserts). Subsequently, we confirmed the expression of adheren β-catenin, which is required for the formation of a functional tight junction (Supplementary Fig. 19). Thereafter, the apical side of the monolayers were challenged with NLCs, free 5-FAM and FAM-SP1 for 24 h. The data showed no detrimental effect of NLCs on TEER and cell layer capacitance and β-catenin distribution (Supplementary Fig. 19). This confirms preservation of the monolayer integrity following 24 h contact with NLCs. The data also shows NLC uptake by hCMEC/D3 monolayers as well as transport of a fraction across the monolayer (Supplementary Fig. 19), presumably arising from endothelial transcytosis. In contrast, neither the fluorophor 5-FAM nor the scrambled peptide showed notable transport across the BBB transwell (Supplementary Fig. 19).

**FAM-CGY complexion with nucleic acids.** Since NLCs at physiological pH are cationic (Fig. 1k), we tested whether FAM-CGY

can form complexes with nucleic acids such as siRNA and deliver them to the cells. We show FAM-CGY can predominantly form spherical assemblies with siRNA (Fig. 3a). The absence of nanofibres (as demonstrated by TEM) indicates that electrostatic interaction between siRNA and CGY could have negatively affected fibre growth, perhaps, by reducing the frequency of R amino acid-mediated bidentate hydrogen bonding. The results in Fig. 3a suggest that the uptake of FAM-CGY/siRNA nanoparticles by hCMEC/D3 cells is predominantly through TfR, since transferrin, but not the RAGE peptide, could suppress uptake. Macropinocytosis, however, may still play a role in the uptake of these nanostructures. When we employed a TfR-specific siRNA in nanocomplex assembly, maximum TfR downregulation was achieved with spherical FAM-CGY/siRNA nanoparticles carrying 24 nM siRNA (corresponding to ~80% TfR down regulation) as demonstrated by Western blot (Fig. 3b). Control experiments with free TfR-specific siRNA and nanocomplexes with a scrambled siRNA did not affect TfR expression. These nano-assemblies were superior in down-regulating TfR expression compared with siRNA delivered with siPORT amine (a commercial transfectant). As an alternative example, we further demonstrated down-regulation of claudin-5 (a trans-membrane tight junction protein highly expressed by brain endothelial cells)[42], through delivery of claudin-5-specific siRNA with FAM-CGY (Supplementary Fig. 20).

With respect to safety, neither FAM-CGY NLCs nor FAM-CGY/siRNA nanoparticles perturbed plasma membrane integrity as demonstrated through measurements of extracellular levels of lactate dehydrogenase (LDH) (Fig. 3c). In addition to this, we also employed an integrated metabolomics approach that measures ATP turnover and mitochondrial oxidative phosphorylation (OXPHOS) by high-resolution real-time respirometry[43]. Calculations of ATP turnover, coupling efficiency of OXPHOS and respiratory control ratio (RCR) showed no significant changes on either FAM-CGY NLC or FAM-CGY/siRNA nanoparticle treatment compared with control (Fig. 3d, Supplementary Table 1). Collectively, these observations confer cell viability and safety, since OXPHOS accounts for the majority of ATP production in cells and mitochondrial dysfunction plays a major role in initiation of many types of cell-death processes[43,44]. These results are in stark contrast to known cytotoxicity of Aβ oligomers[37]. Finally, in contrast to FAM-CGY/siRNA nanocomplexes, siRNA delivery with siPORT was cytotoxic, it triggered LDH release and suppressed ATP turnover, OXPHOS and RCR (Fig. 3c, d, Supplementary Table 1).

For silencing, siRNA must reach the cytoplasmic compartment. Thus, the efficient and selective down regulation of TfR and caludin-5 achieved in our experiments could be indicative of some levels of FAM-CGY/siRNA nanoparticle dismantling in the late endosomes with, concomitant, destabilisation of endosomal

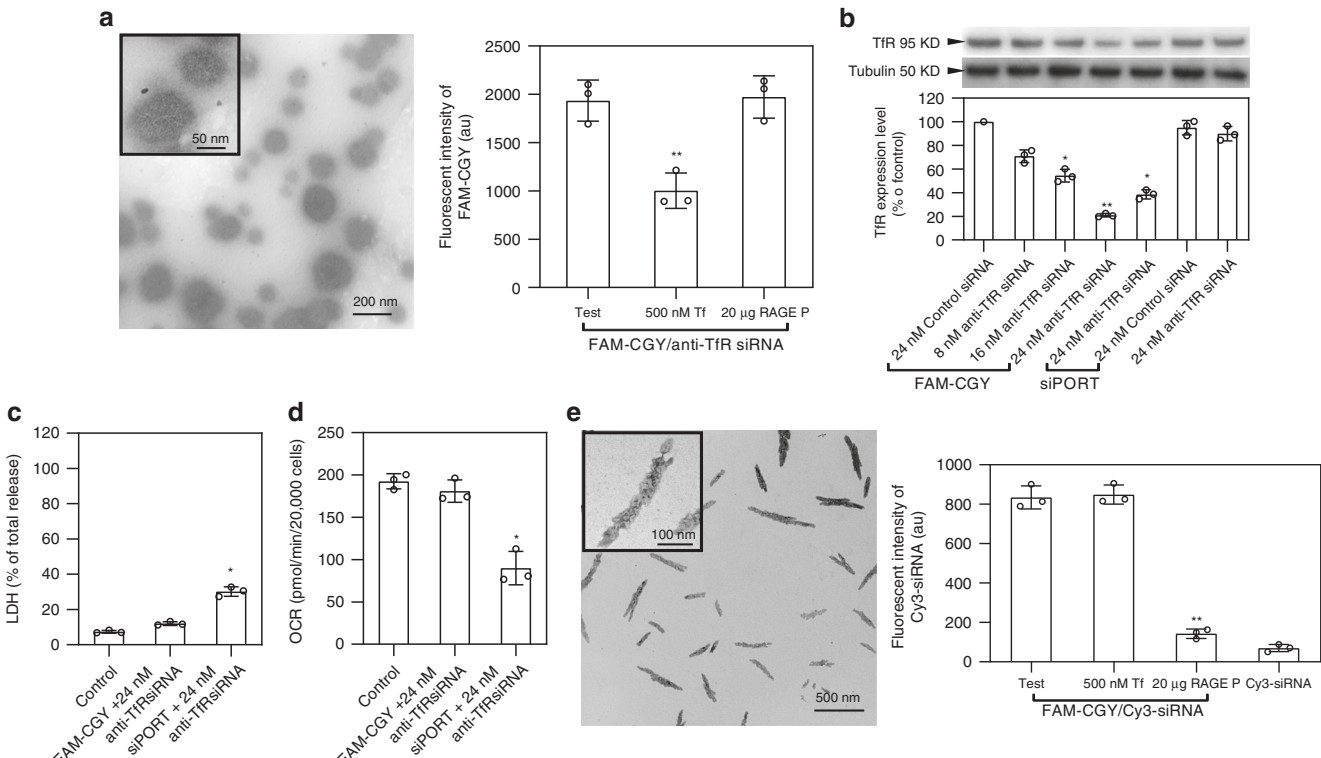

**Fig. 3** NLC-siRNA complexes and their cellular performance. **a** Transmission electron micrograph of FAM-CGY/TfR-specific siRNA complexes (left panel) and their uptake by hCMEC/D3 cells. The uptake is significantly reduced in the presence of human Tf, but not the RAGE P ligand. **b** FAM-CGY/TfR-specific siRNA complexes effectively silence TfR in hCMEC/D3 cells as determined by Western blotting. Silencing is also compared with TfR-specific siRNA alone and when delivered with the commercial transfectant siPORT. **c** Transfection with siPORT, but not with FAM-CGY, causes cell damage as measured through cytoplasmic lactate dehydrogenase (LDH) release. **d** Transfection with FAM-CGY has no significant effect on cell oxygen consumption rate. ATP turnover and respiratory control ratio are unaffected by FAM-CGY (see Supplementary Table 1 for full data analysis). **e** The left panel shows transmission electron micrographs of complexes between FAM-CGY and Cy3-siRNA forming twisted single or multiple fibres. The right panel shows the uptake FAM-CGY/Cy3-siRNA nanofibres (but not naked Cy3-siRNA) by hCMEC/D3 cells, where uptake can be blocked by the RAGE peptide (RAGE P) ligand. Human transferrin (Tf) presence has no effect on fibre uptake. Incubations are in triplicate and each experiment was repeated three times. Panels represent the mean value of three separate experiments ± s.d. and each dot indicates the mean of three technical replicates. *$p < 0.05$ and **$p < 0.01$, non-paired two-sided student $t$-test compared with respective controls. Source data are available in the Source Data file

membrane[45]. NTA studies support instability of FAM-CGY assemblies at low pH values encountered in the late endosomes (Supplementary Fig. 21). NLC dismantling may proceed in late endosome through protonation of H amino acids (contributing to the loss of aromatic stabilisation) and disulfide reduction[46]. In a similar manner to other reported endosome destabilising cationic peptides, cationic FAM-CGY monomers may also express an endosome membrane destabilising property and aid siRNA release into the cytoplasm [45].

Next, we drove a predominant nanofibre assembly formation between FAM-CGY and siRNA. This was achieved by employing a fluorophore labelled siRNA (Cy3-siRNA) to promote π–π interaction with FAM. The resultant nanofibres exhibited an average length of $297 \pm 97$ nm and width of $50 \pm 11$ nm ($n = 100$), respectively (Fig. 3e). These nanofibres provided a further opportunity to discriminate between RAGE- and TfR-dependent uptake mechanisms. To this end, first we showed poor uptake of free Cy3-labelled siRNA by hCMEC/D3 cells, whereas Cy3-siRNA was successfully taken up through nanofibre delivery (Fig. 3e). Competition studies demonstrated that the RAGE peptide ligand, but not transferrin, block FAM-CGY/Cy3-siRNA uptake (Fig. 3e), since cellular Cy3-siRNA levels are comparable to incubations with free Cy3-siRNA. Therefore, RAGE appears to play an important role in the uptake of fibrous NLCs <400 nm and at least in complexion with nucleic acids.

This is in contrast to an earlier study[37], which demonstrated that HeLa cells and a human neuroblastoma cell line could not internalise fibrillar Aβ$_{1-42}$. These differences may be related to considerably longer length of fibrillar Aβ$_{1-42}$ (>1 μm) than NLCs and/or differences in receptor expression/functionality among different cells. Nevertheless, our observations may also explain why previously described amphiphilic peptide fibres had shown some specificity for the brain endothelium[24], as the recognition process might have been mediated through RAGE binding due to its substrate specificity for large size multiple crossed β-sheet assemblies.

**Brain targeting with intravenously injected NLCs.** For intravenous injection and in vivo fluorescent imaging we first used NLCs assembled from Cy5.5-CGY. Intravenously injected NLCs are not only cleared from the blood within the first few hours of injection, but rapidly target the brain and observable within 15 min of injection (Fig. 4a, b). This rapid brain translocation is conceptually consistent with active targeting dogma under sheer flow conditions[3,4]. The peak brain level (after correction for the blood content) is reached at 4 h and corresponds to 5.7% of the injected dose. Thereafter, the results show a decline in the brain NLC level, which may be attributed to local nanoparticle dismantling and/or Cy5.5-CGY degradation. In contrast, Cy5.5-

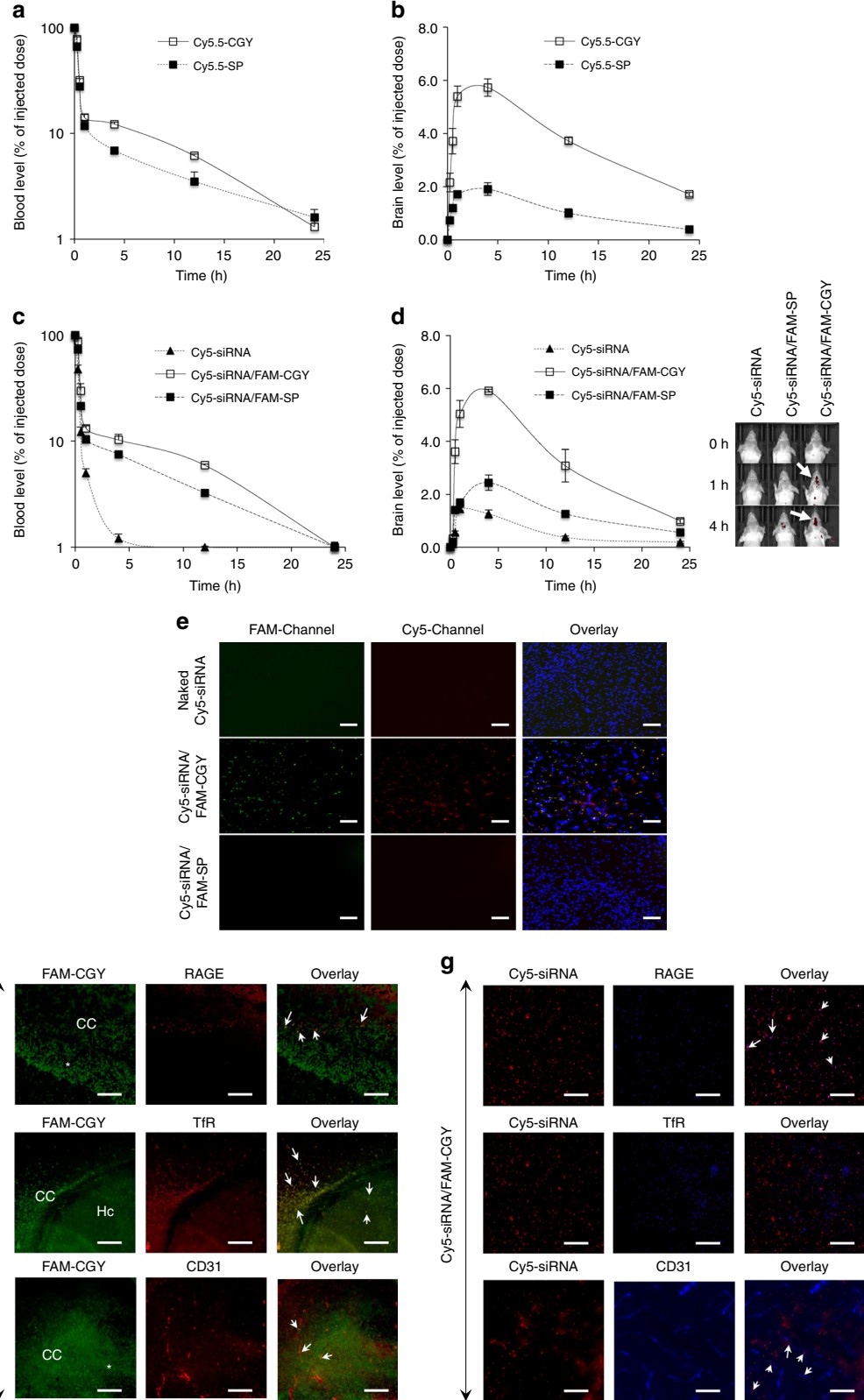

labelled scrambled peptide (Table 1) poorly accumulates in the brain (Fig. 4b) thus confirming FAM-CGY specificity in brain targeting. In contrast to our observations, the best reported active targeting with an optimised transferrin-decorated nanoparticle formulation (a system ascribed with high target binding avidity) has only delivered 1% of intravenously injected dose to the brain at 8 h post injection[12].

Next, we evaluated whether nanofibres formed between FAM-CGY and a Cy5-labelled siRNA can also target the brain and deliver siRNA. The blood clearance profile of nanofibres mirrors that of native NLCs (Fig. 4c). The results also show considerable siRNA deposition to the brain (observable within an hour of injection and peaks by 4 h), which was not achievable on injection of naked siRNA or siRNA in combination with the FAM-labelled

**Fig. 4** Blood clearance and brain deposition level of intravenously injected NLCs in mice. **a** Blood clearance of intravenously injected Cy5.5-CGY and Cy5.5 scrambled peptide (SP). **b** Brain levels of Cy5.5-CGY and its scrambled formulation after correction for the blood content of the brain. **c** Blood clearance of intravenously injected Cy5-siRNA/FAM-CGY complexes compared with naked Cy5-siRNA and CY5-siRNA complexion with FAM-scrambled peptide (SP). **d** Brain levels of Cy5-siRNA delivered through the aforementioned formulations in (**c**). The brain level of siRNA is subtracted from the organ blood level. The right section shows animal imaging of siRNA accumulation in the head region. A typical animal image for each formulation is shown. In all experiments the final concentration of the peptide conjugate was 10 μM and that of siRNA was 48 nM, based on blood volume calculation equivalent to 6% of the body weight. All injections were done in groups of three animals and each experiment was repeated three times. The results are mean values ± s.d. **e** Fluorescent microscopy images of brain sections (following transcardial perfusion and fixation) showing the presence of intact Cy5-siRNA/FAM-CGY in the brain (yellow colour in overlay). **f** Fluorescent microscopy images of the brain sections showing the presence of FAM-CGY in the brain and co-localisation with CD31[+] endothelial cells as well as TfR and RAGE (some examples are depicted with arrows in overlay images). CC = cerebral cortex, Hc = hippocampus and * denotes deep cerebral white matter. **g** Fluorescent microscopy images of the brain sections showing the presence of Cy5-siRNA (delivered through complexion with FAM-CGY) in the brain and co-localisation with CD31 positive endothelial cells and RAGE (some examples are depicted with arrows in overlay images), but not TfR. In both (**f**) and (**g**) NLCs appear in the brain parenchyma. Each experiment was repeated at least three times (in three different animals) with different sections (n = 9). Scale bar = 100 μm. Source data are available in the Source Data file

scrambled peptide (Fig. 4d). Again, the kinetics of siRNA accumulation in the brain parallels that of native NLCs, thus suggesting siRNA delivery is mediated by FAM-CGY. Quantitatively, these results correspond to peak brain siRNA level of 4.03 pmoles at 4 h post injection of FAM-CGY/Cy5-labelled siRNA complexes compared with 0.84 pmole in the case of naked Cy5-siRNA. Naked siRNA is subjected to rapid renal, and to some extent, hepatic clearance[47,48]. Our results further show delay in renal clearance of Cy5-siRNA delivered through complexion with FAM-CGY (siRNA brain to kidney ratio 1:8 and 1:59 for FAM-CGY/Cy5-siRNA and naked siRNA injections, respectively) (Supplementary Fig. 22).

Fluorescent microscopy of the brain sections showed the presence of NLC fluorescent (with and without the fluorescent siRNA cargo) in the brain parenchyma (Fig. 4e). This may be an indicative of active uptake by cerebral endothelial cells and transcytosis through RAGE and TfR[31,49] and consistent with in vitro studies. Accordingly, immunofluorescent brain sections show strong FAM-CGY NLC association with CD31[+] cerebral capillary endothelial cells (Fig. 4f). Furthermore, the NLC-derived fluorescent overlay with both TfR and RAGE in the brain sections. Next, we investigated whether FAM-CGY/Cy5-labelled siRNA nanofibres exclusively overlay with RAGE. Indeed, the results not only show some overlay between Cy5-siRNA and CD31[+] endothelial cells, but also with RAGE (Fig. 4g). Here, no overlay was observed with TfR. Therefore, these observations are in accord with in vitro studies, which showed nanofibre uptake by cerebral endothelial cells is predominantly through RAGE recognition.

Since NLCs appear in the brain parenchyma we further examined whether they could be taken up by different parenchymal cells. The results in Fig. 5a, b show association of both FAM-CGY and FAM-CGY/Cy5-siRNA NLCs with neurons and microglial cells, but not with astrocytes. Neurons and microglial cells express TfR and RAGE[50,51] and NLC uptake by these cells may have been through these receptors, but this was not investigated. The lack of NLC uptake by astrocytes is interesting, despite indications that astrocytes also express TfR and RAGE[50,51]. The reasons for these difference remains unknown and may be related to the extent of receptor expression and functionality among these cells (and microenvironmental control), but it is also plausible that trans-endothelial transport processes may have modified NLC properties, presenting them as substrates for neurons and microglial cells.

Next, we assessed whether NLC/siRNA complexes can exert pharmacological activity in the brain. This was performed by measuring extent of β-secretase 1 (BACE1) down regulation in hippocampus following intravenous injection of FAM-CGY/β-

secretase 1 (BACE1)-specific siRNA complexes. BACE1 is highly expressed in neurons and it is responsible for initiating Aβ generation and therefore has been considered as an important target for the therapeutic inhibition of Aβ production in Alzheimer's disease[52]. The results show that a single intravenous injection of FAM-CGY/BACE1-siRNA nanoparticles down regulates BACE1 expression by ∼50% compared with FAM-CGY/control siRNA complexes as determined by Western blot (Fig. 5c). Furthermore, intravenous injection of BACE1 siRNA either in free form or with FAM-SP1 showed no pharmacological activity in hippocampus. These observations provide a pharmacological proof-of-concept for functionality of a nucleic acid medicine in the brain through NLC delivery and validate the capability of the engineered NLCs to functional as neurological nanomedicines.

**NLC safety**. We studied the effect of intravenously injected FAM-CGY, FAM-CGY/BACE1-specific siRNA (spheroidal objects) and FAM-CGY/Cy5-siRNA (nanofibres) NLCs on the structure of all main organs. The haematoxylin and eosin sections of the brain (cerebral cortex and hippocampus) show that NLC treatment does not induce neuronal injury/swelling, cellular liquefaction, necrosis and focal inflammatory cellular infiltration (Fig. 6a–d, Supplementary Fig. 23). In addition to these observations, biochemical analysis of brain homogenates for inflammatory markers tumour necrosis factor-α (TNF-α), interleukin-1β (IL-1β), IL-6, ionised calcium-binding adaptor molecule 1 (Iba-1; microglia marker) and glial fibrillary acid protein (GFAP; astrocyte marker) also revealed no inflammation in response to NLC treatment (Fig. 6e–i).

In line with the brain investigations, no adverse morphological changes and inflammatory reactions were observable throughout the liver and the kidneys (including renal cortex, renal corpuscle, glomerulus, proximal convoluted tubule, juxtaglomerular apparatus and renal medulla) (Supplementary Fig. 24) as well as other major organs (heart, lungs and spleen) (Supplementary Fig. 25) at all studied time points and the results were similar to those of control animals. These observations are particularly important with respect to the liver and the kidney, as these organs play a key role in the overall NLC extraction from the blood. Furthermore, NLC treatment had no effect on blood count (Supplementary Table 2). As a positive control we show selected organ damage and alterations in blood count on lipopolysaccharide injection. Finally, NLCs did not induce complement activation in human sera (Supplementary Fig. 26). Collectively, these observations together with lack of FAM-CGY (with and without nucleic acids) cytotoxicity in hCMEC/D3 cells confer NLC safety.

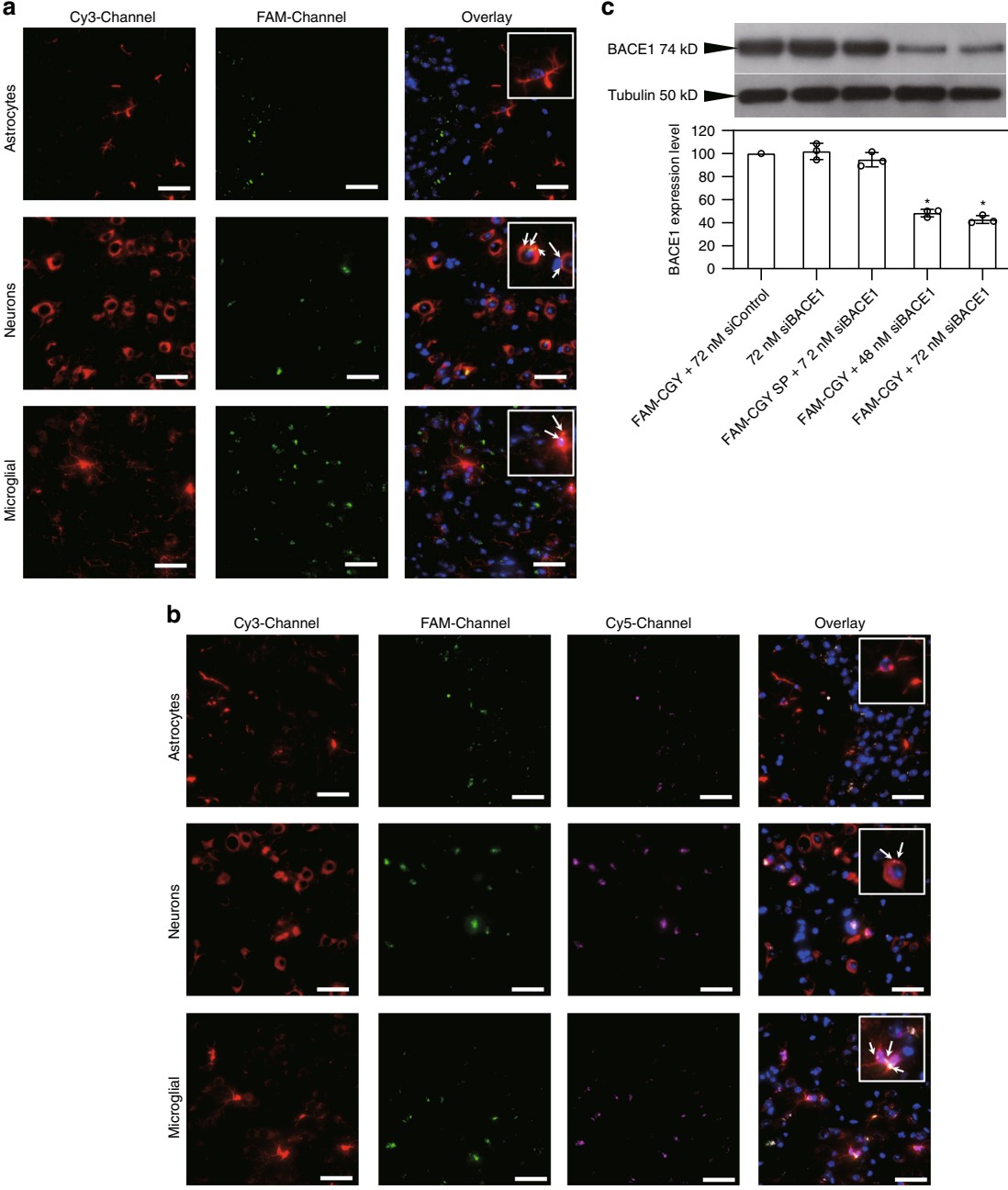

**Fig. 5** Brain localization of NLCs and NLC-siRNA complexes and demonstration of pharmacological activity. **a**, **b** Fluorescent microscopy images of brain cerebral sections at 4 h after the administration of FAM-CGY NLCs (final concentration of 10 μM) (**a**) and FAM-CGY/Cy5-siRNA complexes (FAM-CGY = 10 μM, Cy5-siRNA = 48 nM) (**b**), respectively. Astrocytes, neurons and microglia were labelled with anti-GFAP, anti-Tuj1, anti-Iba1 antibodies, respectively and visualised with a secondary fluorescent antibody (Cy-3 channel). Inserts represent enlarged sections. Arrows indicate FAM-CGY NLC and FAM-CGY/ Cy5-siRNA complexes. Cy3-Channel: astrocytes or neurons or microglial cells; FAM-Channel: FAM-CGY NLCs, Cy5-Channel: Cy5-siRNA. Cell nuclei are stained with DAPI. Scale bar = 40 μm. The images show NLC localization to neurons and microglial cells, but not astrocytes. **c** Western blot demonstration of suppression of $\beta$-secretase 1 (BACE1) expression in hippocampus following a single intravenous injection of FAM-CGY/BACE1 siRNA complexes. Silencing was compared with FAM-CGY/siControl injection. BACE1-specific siRNA alone or delivered with the FAM-CGY scrambled peptide 1 (FAM-CGY SP) did not induce silencing. BACE1 expression was measured 48 h after administration. The results represent mean values of three separate experiments ± s.d. *$p < 0.05$, non-paired two-sided student $t$-test compared with respective siControl (control siRNA). Source data are available in the Source Data file

## Discussion

Through simple chemical modification of the phage-derived GYR peptide we engineered two distinct hierarchical platforms in the form of core-shell nanoparticles and nanofibres capable of targeting at least two distinct receptors (TfR and RAGE) on cerebral capillary endothelial cells. On intravenous injection, these platforms crossed the BBB and reached neurons and microglial cells. Subsequently, we demonstrated siRNA delivery to the brain and showed pharmacological activity through BACE1 silencing. Thus, these attempts have overcome previous limitations in active-targeting with GYR functionalised drug carriers[8,11] and are advantageous over other nano-based approaches, since it neither

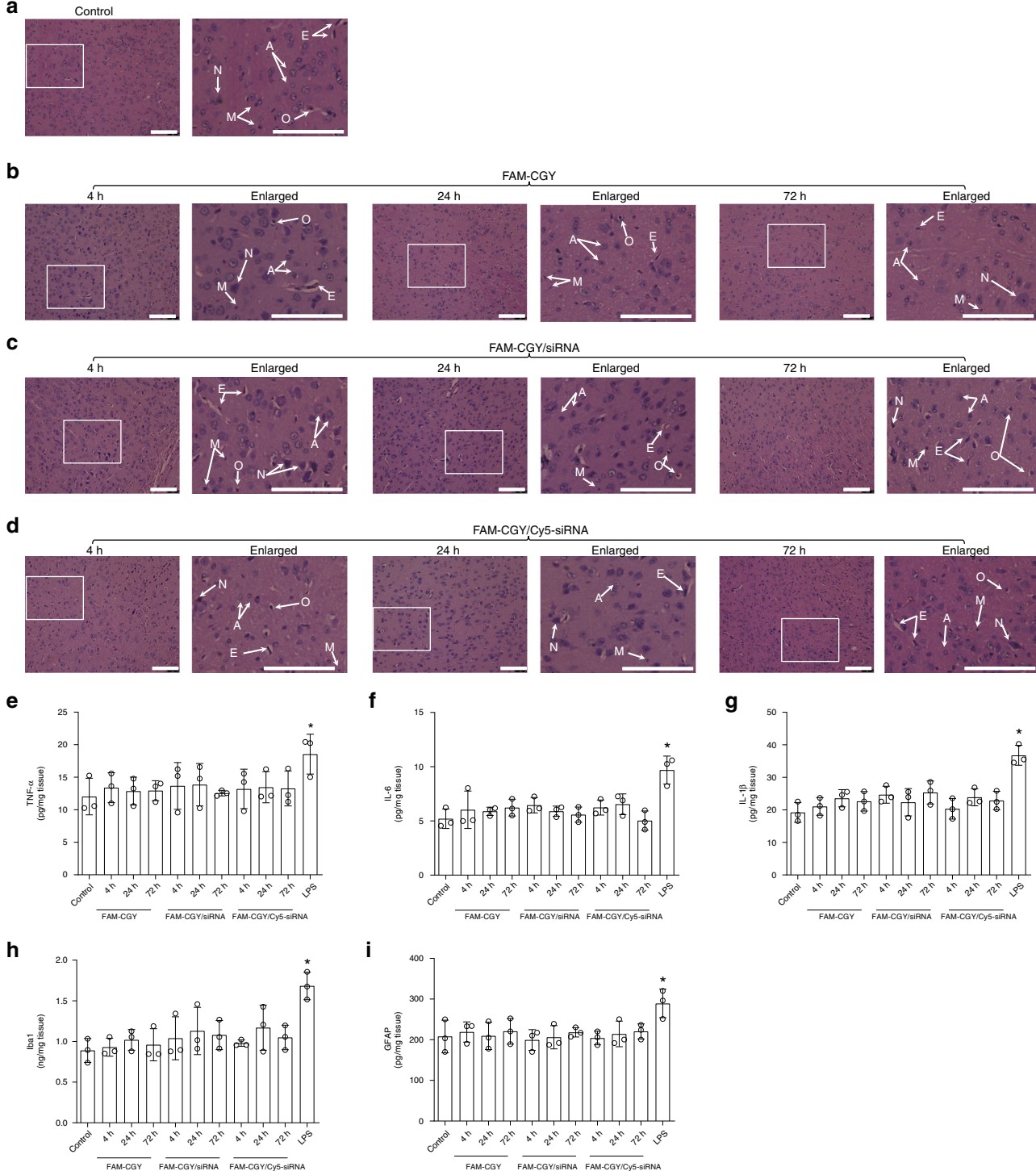

**Fig. 6** Brain safety of intravenously injected NLCs and NLC-siRNA complexes. The images show haematoxylin and eosin stained sections of cerebral cortex at 4 h, 24 h and 72 h after administration of saline (**a**), FAM-CGY peptide (10 μM) (**b**), FAM-CGY/BACE1-specific siRNA (**c**) and FAM-CGY/Cy5-siRNA complexes (**d**). In (**c**) and (**d**) FAM-CGY and siRNA concentrations were 10 μM and 48 nM, respectively. Magnified regions of box inserts are shown. E = cerebral endothelial cells, G = glial cells, M = microglial cells and N = Neurons. Scale bar = 100 μm. ELISA determination of pro-inflammatory markers TNF-α (**e**), IL-6 (**f**) and IL-1β (**g**) in hippocampus on NLC and NLC-siRNA treatments compared with control ($p > 0.05$ in all cases; $n = 3$ determinations). Determination of microglial marker Iba-1 (**h**) and astrocyte marker GFAP (**i**) in hippocampus on NLC treatment ($p > 0.05$ in all cases; $n = 3$ determinations). Lipopolysaccharide (LPS) injection induced a small, but significant (*$p < 0.05$ in all cases compared with control and NLC treatments; $n = 3$ determinations) proinflammatory response at 24 h sacrifice point. All statistical analyses were performed with one-way ANOVA, using Tukey's multiple comparison correction to calculate significance. Source data are available in the Source Data file

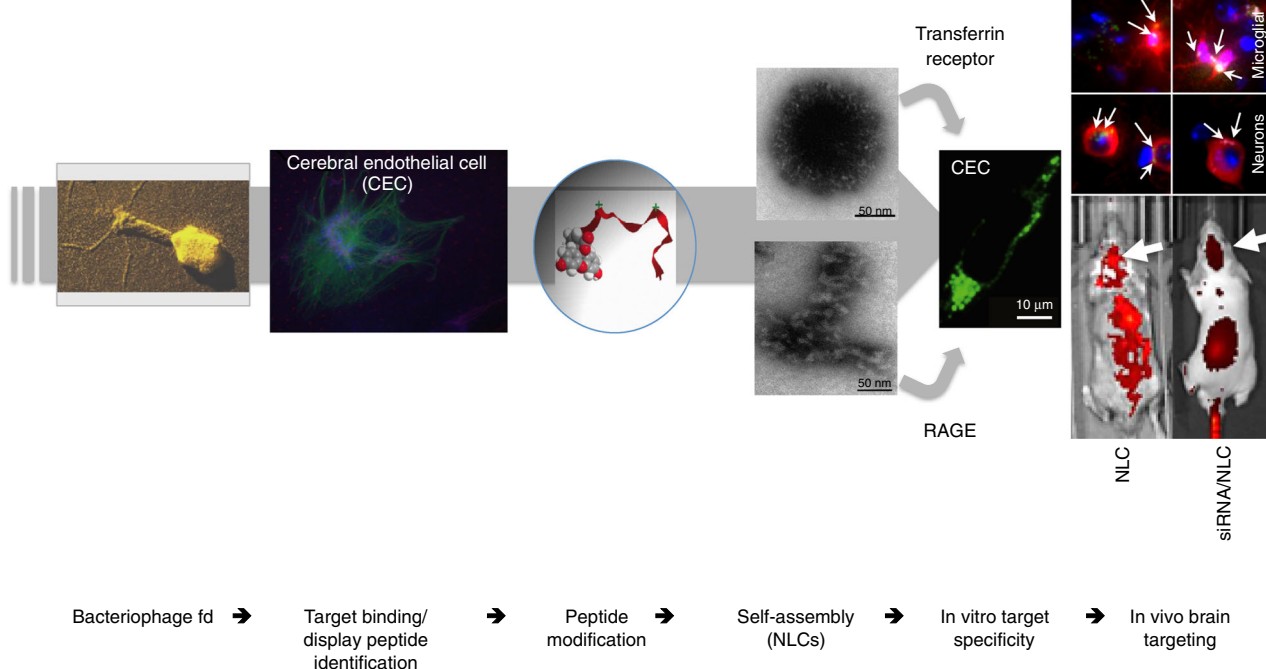

**Fig. 7** Schematic summary of NLC construction and active targeting of the brain. Animal images represent FAM-CGY NLC (left; corresponding to core-shell nanoparticle morphology) and Cy5-siRNA/FAM-CGY NLC (right; corresponding to nanofibre morphology) deposition in the brain at 2 h post intravenous injection and corresponding immunofluorescent brain sections at 4 h, showing NLC deposition in neurons and microglial cells

require complex design nor extensive metabolic modulation/perturbation in the whole organism (e.g., glucose deprivation)[53].

At large, active translocation of hydrophobe-CGY/nucleic acid complexes into the brain opens many opportunities for genetic manipulation of the blood-brain-barrier and parenchymal cells involved in escalation of various neurodegenerative diseases[54]. Importantly, unlike intracellular Aβ[37], NLCs do not induce bioenergetics crisis and cell death and therefore are perceived as effective and safe delivery systems (or nanotherapeutics).

In summary, we have provided a conceptual framework in active targeting through self-assembly of a peptide ligand into targetable nanocarriers (Fig. 7). Unlike most phage virions[5], NLCs are internalised by target cells, which presumably is a function of their polyvalent surface patterned nature and hence high target binding avidity. Furthermore, NLCs overcome the lack of cell specificity in targeting with cell-penetrating peptides[55]. We foresee future development of multifunctional NLCs with broader specificities, for instance, through replacement of the fluorophore component with other hydrophobes or drugs and/or cross-linking of hydrophobe-CGY conjugates to pharmacological peptides through a disulfide bridge.

## Methods

**Reagents**. For all reagents catalogue numbers (♯number) are indicated. Human transferrin (#10652202001), bovine serum albumin (BSA) (#A9418), pyrene (#82648), crystal violet solution (#HT90132), (tris)2-chloroethy)phosphate (TCEP) (#07296), 2-deoxy-D-glucose (#D8375), sodium azide (#71289), worthmanin (#W1628), chlorpromazine (#C8138), N-ethylmaleimide (#E1271), indomethacin (#I7378) and lipopolysaccharide from Escherichia coli O55:B5 (#L2880) were purchased from Sigma-Aldrich Co. LLC (Seelze, Germany). Texas Red®-Transferrin (#T2875), mouse anti-human transferrin receptor antibody (#A-11130), mouse anti-β-tubulin (#32-2600), HRP-goat-anti-mouse IgG (H+L) (#62-6520), HRP-goat-anti-rabbit IgG (H+L) (#65-6120) and Novex® ECL Chemiluminescent Substrate Reagent Kit (#WP20005) were purchased from Life Technologies (CA, USA). RAGE peptide (#ab32414), rabbit polyclonal to mouse and human RAGE (#ab3611), rabbit anti-mouse and human TfR antibody (#ab84036), rabbit anti-human claudin-5 antibody (#ab131259), anti-Tuj1 antibody (#ab78078, Abcam, UK), anti-GFAP antibody (#ab10062), mouse monoclonal to CD31 (#ab24590, reacts with mouse and human), rabbit polyclonal to CD31 (#ab28364, reacts with

mouse and human), rabbit polyclonal to β-catenin (#ab16051, reacts with human), rabbit monoclonal anti-BACE1 antibody (#ab108394), donkey anti-mouse IgG H&L (Alexa Fluor®680) (#ab175774), donkey anti-mouse IgG H&L (Alexa Fluor®450) (#ab175658) and donkey anti-mouse IgG H&L AlexaFluor555 (#ab150110) were purchased from Abcam (Cambridge, UK). Anti-Iba1 antibody (#ab019-19741) was from Wako, Japan. Amyloid β₁₋₄₂ (Aβ) peptide (#AS-20276) and tetramethylrhodamine (TAMRA)-labelled amyloid β₁₋₄₂ peptide (#AS-60488) were purchase from AnaSpecas (CA, USA). Goat anti-rabbit IgG (H + L) Super-clonal secondary antibody Alexa Fluor 555 (#A27039) was from ThermoFisher Scientific UK. Cy3-siRNA (#AM4621), anti-TfR siRNA (#4390824, siRNA ID: s727), anti-RAGE siRNA (#4392420, siRNA ID: s1166), anti-claudin-5 siRNA (#4392420, siRNA ID: s14245) and siPORT Amine Transfection agent (#AM4503) were purchased from Ambion Inc. (TX, USA). Cy5-siRNA (#siL0727105040) and BACE1 siRNA (sense of 5'-GCUUUGUGGAGAUGGUGGATT-3' and antisense: 5'-UCCACCAUCUCCACAAAGCTT-3', #siB170613094224) were from Guangzhou RiboBio Co., Ltd. (Guangzhou, China). Western blot stripping buffer (#SC-281698) and control siRNA (#sc-37007, sequence not disclosed by the manufacturer) were from Santa Cruz Biotech Inc. (CA, USA). ProteoJET™ Mammalian Cell Lysis Reagent (#K0311) was purchased from Fermentals Life Sciences (Ontario, Canada). Hoechest 33342 (350/461) (#H3569), CellLight® Reagents BacMam 2.0 Early endosomes-RFP (555/584) (#C10587) and Lysosomes-RFP (555/584) (#C10597) were purchased from Molecular Probes, Life Technologies (CA, USA).

**Cell lines and media**. The human brain endothelial cell line (hCMEC/D3) was obtained under licence from University Paris 05, CNRS, Institute Cochin, INSERM (Paris, France). The cell line was maintained and characterised in accordance to regularly updated protocols by Institute Cochin and certified mycoplasma free[56]. Human breast cancer cell line (MCF-7) (ATCC® HTB-22™) and human mammary epithelial cell line (MCF-10A) (ATCC® CRL-10317™) were purchased from American Type Culture Collection (VA, USA). Endothelial Basal Medium (EBM-2) (#CC-3156) was from Lonza Group Ltd. (Basel, Switzerland). Dulbecco/Vogt Modified Eagle's Minimal Essential Medium (DMEM) (#D0819), basic fibroblast growth factor (#F0291), gelatin solution (#G1393), hydrocortisone (#H-0888), insulin (#I-1882) and phosphate buffered saline (PBS) (#D8537) were purchased from Sigma-Aldrich Co. LLC. (Seelze, Germany). Foetal bovine serum (FBS) (#16000044) and penicillin/streptomycin (#15140122) were obtained from Gibco-BRL, Carlsbad Life Technologies (CA, USA). Recombinant human EGF (AF-100-15) was from PeproTech EC Ltd. (Hamburg, Germany). HEPES (#S11-001) was from PAA Cell Culture Company (Pasching, Austria). The cell culture flasks and plates were purchased from Corning Inc. (NY, USA).

**Elisa kits**. Enzyme-linked immunosorbent assay (ELISA) kits for mouse TNF-α (#RAB0477 Millipore), mouse IL-6 (#RAB0309), mouse IL-1β (#RAB0275) and

GFAP (#NS830 Millipore) were supplied by Merck (USA). The ELISA kit for quantitative detection of mouse IBA1 (#LS-F7666) was provided by Lifespan Biosciences Inc. (WA, USA). Elisa kits for quantitative detection of human complement products C3a (#031), C5a (#020) and sC5b-9 (#020) were from Quidel (CA, USA).

**Peptide synthesis.** Peptide synthesis (sequences as in Table 1) was carried out by a solid-phase method as described in detail in Supplementary File (Supplementary Fig. 1, 4–11). Peptides were purified by preparative HPLC (Shimadzu LC-8A) and characterised by the analytical HPLC (Shimadzu LC-20AB) and mass spectroscopy (Shimadzu LCMS2020). The lyophilized peptides were stored at $-20\,°C$ under nitrogen atmosphere.

**Self-assembly.** The critical aggregation concentration of FAM-CGY was determined in Milli-Q (MQ) water using pyrene as a probe ($n = 3$ determinations) at room temperature[57]. SDS-PAGE was carried out using XCell Surelock$^{TM}$ system (Invitrogen, CA, USA) with precast 12 % NuPAGE® Bis-tris mini gels (Invitrogen, CA, USA) operated at a constant voltage of 200 V in NuPAGE® MES running buffer (Invitrogen, CA, USA). For non-reducing conditions, various concentrations of the peptides were prepared by dilution of the stock solution in MQ water and incubated for 30 min at room temperature. The NuPAGE® LDS sample buffer (4×) (Invitrogen, CA, USA) was added to the peptide solutions and 10 µL of each mixture solution was loaded into individual wells. For the reducing condition TCEP was co-incubated with peptide overnight to disrupt the disulfide bonds. Gels were stained with SilverXpress® Silver Staining Kit (Life Technologies, CA, USA) according to the manufacture's protocol. SDS-PAGE experiments were repeated with three different batches of peptides and each experiment as performed at least three times.

**Circular dichroism (CD).** The CD spectrum of CGY, FAM-CGY peptide (at concentrations below and above the CAC) was measured on a Jasco J-810 spectropolarimeter (Jacso Incorporated, MD, USA) with 4 s accumulations every 1 nm and averaged over three acquisitions.

**Microscopy.** For AFM, a sample of 5 µM FAM-CGY (or other analogues and complexes thereof) was applied to a cleaned mica surface and air-dried at room temperature for 30 min. AFM was performed on a tapping mode with a commercial Nanoscope IV MultiMode SPM (Veeco, Santa Barbara, USA). A 12 µm piezoscanner (E scanner) (Veeco, Santa Barbara, USA) was employed for imaging. For TEM, FAM-CGY (or other analogues and complexes) was negatively stained in accordance with an established procedure. Briefly, a 10 µL sample of 5 µM FAM-CGY (or other complexes) was placed onto a 200 mesh carbon-coated copper grids and allowed to stand for 5 min. Excess solvent was carefully removed by capillary action using a filter paper and the sample was immediately stained with 10 µL of 2% (v/v) phosphotungstic acid solution for 2 min. Excess stain was removed and the grids were allowed to dry for 20 min. Images were taken with a Philips CM100 transmission electron microscope (FEI/Philips, Eindhoven, Netherland) with an accelerating voltage of 80 kV. Nanoparticle dimensions were determined using Image J software (http://rsb.info.nih.gov/ij/download.html) and reported as the mean size of at least 100 randomly selected images ± s.d.

**Nanoparticle Tracking Analysis (NTA).** NTA measurements were performed with a NanoSight LM20 (NanoSight, Amesbury, United Kingdom) equipped with a sample chamber with a 405 nm blue laser and a Viton fluoroelastomer O-ring[57]. Briefly, different concentrations of peptides and complexes were prepared and diluted appropriately before measurement. Size and particle concentration measurements were performed at room temperature as well as at 37 °C. Each experiment was repeated at least three times and the results are presented as mean ± s.d. When necessary, a typical distribution profile is shown.

**Zeta potential.** The zeta potential of FAM-CGY and other complexes was calculated from electrophoretic mobility measurements using Zetasizer Nano ZS (Malvern, UK) at 21 °C. Each experiment was repeated at least three times and the results are presented as mean ± s.d.

**Cell culture.** hCMEC/D3, MCF-7 and MCF-10A cells were grown either per manufacturer's instructions or following the guideline of the donated laboratory. Briefly, hCMEC/D3 cells were grown in EBM-2 medium supplemented with 5% (v/v) FBS, 1.4 µM hydrocortisone, 1 ng mL$^{-1}$ basic fibroblast growth factor, 1% (w/v) penicillin/streptomycin and 10 nM HEPES[58]. MCF-7 and MCF-10A cells were grown in DMEM containing 10% (v/v) FBS, 1% (w/v) penicillin/streptomycin, 0.5 µg mL$^{-1}$ hydrocortisone, 10 µg mL$^{-1}$ insulin and 20 ng mL$^{-1}$ recombinant human EGF. All cells were maintained at 37 °C in a humidified atmosphere. Adherent cells were harvested with 0.05% (w/v) trypsin at a sub-cultivation ratio of 1:3, while non-adherent cells were seeded at a density of $7 \times 10^5$ cells per flask in 75-Flask and maintained at 70–80% confluence.

**Cell uptake and trafficking.** Peptide internalisation was measured by flow cytometry (FACS ArrayTM Cell Analysis, Becton, Dickinson and Company, NJ, USA). Cells ($1 \times 10^5$ cm$^{-2}$) were seeded on 24-well plates and grown 1 day at 37 °C and 5% CO$_2$ in order to reach 60–70% confluence. The cells were washed 3 times with pre-heated PBS. The uptake studies were initiated by adding 500 µL of 5 µM fluorescence-labelled peptides (diluted in cell medium containing with different serum concentration). Serum concentration in incubation varied from 5 to 20% (v/v). After 24 h of incubation, treated cells were then washed three times with pre-warmed PBS and harvested by trypsinisation. A total of 10,000 cells were analysed by flow cytometry.

Intracellular trafficking of NLCs was monitored by live-cell fluorescent microscopy. Briefly, hCMEC/D3 cells ($2 \times 10^4$ cm$^{-2}$) were seeded on the 8-well Lab-Tek chamber slides (Nunc, IL, USA) for 1 day at 37 °C and 5% CO$_2$ to reach 60–70% confluence. CellLight® Reagents BacMam 2.0 Early endosomes-RFP, or Lysosomes-RFP was added to cells and incubated for 24 h[58]. The cells were washed three times with pre-heated PBS and the uptake studies were initiated by adding 200 µL samples of different NLCs (diluted in cell medium and incubated for 30 min at 37 °C prior to addition). After 4 h or 24 h of incubation at 37 °C, each chamber was washed three times with pre-heated PBS and incubated with the respective fresh cell growth medium. The cell nucleus was stained with Hoechst 34580 dye (5 µg mL$^{-1}$)[58,59]. Live-cell imaging was performed on a wide-field microscope (Leica AF6000LX, Hamburg Germany) using a 63× oil objective with 1.6 magnification and analysed with appropriate filters. Z-stacking was performed using appropriate sectioning steps ranging from 0.2 to 0.7 µm. Diffraction PSF 3D was used to calculate the point spread function followed by 3D deconvolution by Leica LAS AF Lite software[58,59]. The co-localisation analysis was processed with Image J to calculate Manders' coefficient.

In some experiments, cells ($2 \times 10^4$ cm$^{-2}$) were treated with a range of internalisation inhibitors at sub-cytotoxic concentrations (confirmed by Trypan Blue exclusion tests where cell viability >95%) for 1 h at 37 °C. The medium was then removed and replaced with a fresh medium containing NLCs and the corresponding inhibitor. After 8 h of incubation at 37 °C, cells were washed three times with cell medium and analysed by FACS. Cell viability at the end of each experiment was >90% as determined by Trypan Blue exclusion test. The following inhibitors were used: 2-deoxy-D-glucose (1 mM)/sodium azide (1 mM) as energy-dependent inhibitors, worthmanin (10 µM) as macropinocytosis inhibitor, chloropromazine (30 µM) as clathrin-dependent endocytosis inhibitor, N-ethylmaleimide (5 µM) as caveolae-dependent endocytosis and transcytosis inhibitor and indomethacin (200 µM) as caveolae-dependent endocytosis inhibitor.

**Competition studies.** In competition experiments, 5 µM NLCs were co-incubated with different concentrations of transferrin or RAGE peptide or amyloid $β_{1-42}$ oligomers in hCMEC/D3 cells for 16 h. The amyloid $β_{1-42}$ oligomers were prepared using a previous method[60]. BSA was used as control protein competitor. The fluorescence microscopy and FACS were employed to observe and quantify FAM-CGY self-assembly uptake, respectively.

In some experiments, MCF-7 cells ($5 \times 10^5$ per well) were incubated in Dulbecco's phosphate buffered saline (DPBS) supplemented with 1% w/w BSA and 0.1% (w/w) sodium azide for 30 min at 4 °C. Then after, incubations were continued for a further 1 h with increasing concentrations of holo transferrin. Next, cells were washed three times with DPBS and then incubated with 5 µM FAM-CGY NLCs in DPBS [supplemented with 1% (w/w) BSA and 0.1% (w/w) sodium azide] for 1 h at 4 °C. Finally, cells were washed three times and resuspended in ice cold cell media for FACS analysis.

**Silencing.** hCMEC/D3 cells were transfected with anti-TfR siRNA [sense sequence (5′-GGUCAUCAGGAUUGCCUAAtt-3′) and antisense sequence (5′-UUAGGCA AUCCUGAUGACCga-3′)] or anti-RAGE siRNA [sense sequence (5′-GGUGGAA CCGUAACCUGAtt-3′) and antisense sequence 5′-UCAGGGUUACGGUUCCAC Cag-3′] or control siRNA using siPORT Amine transfection agent, according to the manufacturer' Neofection protocol. Briefly, siPORT Amine transfection reagent (7 µL) was added to serum-free cell medium to a final volume of 100 µL and vortexed. Next, 70 pmol siRNA in 100 µL serum-free cell medium was added dropwise to the diluted siPORT Amine transfection reagent. The mixture was gently mixed and incubated at room temperature for 20 min. The transfection agent/siRNA complex was added to culture plates containing $2.3 \times 10^5$ hCMEC/D3 cells in 2.3 mL normal growth medium. The final concentration of siRNA was 28 nM. After 24 h incubation, the medium containing transfection agent/siRNA complex was removed and fresh growth medium was added. After 48 h post transfection, the level of target protein (TfR or RAGE) was determined by Western blotting. The hCMEC/D3 cells that were transfected with anti-TfR siRNA or anti-RAGE siRNA or control siRNA for 72 h, were washed with PBS three times and then incubated with 5 µM FAM-CGY nano-assemblies, 62.5 nM Texas Red-labelled transferrin (TfR substrate) and TAMRA-labelled amyloid $β_{1-42}$ (RAGE substrate). After 16 h of incubation at 37 °C, cells were washed three times with pre-warmed PBS and analysed by fluorescence microscopy. Uptake was quantified by measuring fluorescence using FACS.

**Receptor expression**. Transferrin receptor and RAGE expression level in human cells was determined by Western blot analysis. Briefly, cells were seeded in 75 cm²-flasks and grown to reach 60–70% confluence. The cells were washed in ice-cold PBS three times and then scraped off the flask and collected by centrifugation. The pelleted cells were lysed for 10 min at room temperature on a shaker (900 rpm) following the ProteoJET™ Mammalian Cell Lysis Reagent protocol. Lysates were spun in a centrifuge at $18,000 \times g$ for 15 min and the supernatant was collected. Equal protein aliquots were resolved by SDS-PAGE, transferred to nitrocellulose membranes using iBot Dry Blotting system (Invitrogen, CA, USA), immunoblotted with primary antibodies (mouse anti-human TfR antibody or rabbit polyclonal to mouse and human RAGE) (1:200 and 1:750, respectively) and detected with HRP-Goat-anti-mouse (or rabbit) IgG (H + L) secondary antibody (1:3000). Tubulin was immunoblotted with mouse anti-human-β-tubulin (1:200) and detected with HRP-Goat-anti-mouse IgG (H + L) secondary antibody (1:3000). These were followed by incubation with Novex® ECL Chemiluminescent Substrate Reagent Kit. The Bands were quantified with ImageJ 1.44p (http://imagej.nih.gov/ij/).

**FAM-CGY/siRNA assemblies**. Peptide/siRNA assemblies were formed by dropwise addition (80–240 nM range) of either Cy3-siRNA or functional anti-TfR siRNA or anti-Claudin-5 siRNA (with the sense and antisense sequences of 5′-CCUUAACAGACGGAAUGAAtt-3′ and 5′-UUCAUUCCGUCUGUUAAGGgc-3′) to 50 μM FAM-CGY in physiological saline and incubated for 30 min. Next, the mixtures were diluted with saline to a final peptide concentration 5 μM for TEM studies. In vitro uptake and silencing experiments were performed in hCMEC/D3 cells. Briefly, peptide/siRNA nano-assemblies (5 μM FAM-CGY and either 8 or 16 or 24 nM functional siRNA, final concentration) were added to $2.3 \times 10^5$ hCMEC/D3 cells. After 24 h incubation, the medium was replaced by fresh growth medium. After another 48 h of incubation, the level of the target protein was determined by Western blotting. The results were compared with parallel experiments containing nano-assemblies formed from FAM-CGY and an irrelevant siRNA as well as transfection procedures with complexes formed between siRNA and siPORT Amine transfection agent. Cy3-siRNA uptake was quantified by measuring median cell fluorescence using FACS.

**Cell functionality and viability**. LDH release was followed at 24 h post transfection procedures. The measurement was performed using CytoTox96® Non-Radioactive Cytotoxicity Assay kit (Promega, WI, USA)[43]. The maximum amount of LDH in the cells, induced by the addition of a lysis solution, was measured and used as a 100% LDH release and compared with peptidoplex and siPORT-siRNA complex-induced LDH release as well as to spontaneous cellular LDH release (untreated cells). To investigate the possible adverse effects of transfectants on cell respiration, hCMEC/D3 cells were seeded in XF96 V3 cell culture microplates (Seahorse Bioscience, CA, USA) at $1.0 \times 10^4$ cells per well in growth medium. The day after, cells were incubated with designated concentrations of transfectants at 37 °C and 5% $CO_2$ for 24 h. Following incubation, medium was replaced with serum and bicarbonate free assay medium (Seahorse Bioscience, CA, USA) 30 min before monitoring the oxygen consumption rate (OCR) in real-time using XF96 Analyzer (Seahorse Bioscience CA, USA)[43,61]. Different respiratory states were analysed in order to calculate the coupling efficiency of OXPHOS and the mitochondrial RCR[43,61]. Data was corrected for any possible effect of difference in cell numbers[41,56]. Cell numbers were evaluated by growing XF96 V3 cell culture microplate in parallel and following incubation with designated concentrations of transfectants. Cells were fixed with 11% (v/v) glyceraldehyde and stained with crystal violet. Crystal violet was then extracted with 10% (v/v) acetic acid and the absorbance measured at $\lambda = 595$ nm. Standard curve was made to demonstrate the linear relationship between cell numbers and crystal violet staining and for obtaining accurate cell numbers. The real-time OCR was thereafter normalised for cell numbers using the absorbance values[43,61].

**Transwell studies**. For the construction of the in vitro BBB model, hCMEC/D3 cells were seeded on rat tail collagen-coated polycarbonate membranes (Transwell, No. 3401 Costar; Corning, Wiesbaden, Germany; 0.4 μm pore size; 1.13 cm² growth area) with a density of 100,000 cells cm⁻². Every three days the cell medium was change. In general after 7 days growth, the cells reached confluence. The transendothelial electrical resistance (TEER) and cell layer capacitance ($C_{CL}$), which reflects the membrane surface area of the hCMEC/D3 cell monolayer, were measured using CellZscope system (NanoAnalytics, Münster, Germany). Cells with $C_{CL}$ values in the range of 0.5–5.0 μF cm⁻² indicate cell confluence and validate TEER values were selected for the transfer experiments. NLCs at different concentrations were introduced in to the apical side of the BBB model. The TEER values were measured during the incubation. After 24 h of incubation, the media from basal and apical sides were sampled and analysed for fluorescent measurement. CD31 and paracellular adherens (β-catenin) junctions were assessed by immunofluorescence labeling directly on the cell monolayer attached to the transwell membrane. Briefly, cells were fixed by 4% v/v paraformaldehyde for 10 min, rinsed and then permeabilised with 0.5% (v/v) Triton X-100 for 10 min. After rinsing samples were blocked by 5% w/v BSA for 1 h, followed by incubation with primary antibodies (rabbit polyclonal to CD31, rabbit polyclonal to anti-β-catenin) in 1% w/v BSA overnight at 4 °C. Next, samples were rinsed and incubated with

10 μg mL⁻¹ secondary antibody (Alexa Fluor® 555 goat anti-rabbit IgG, Life Technologies, Eugene, OR, USA) in 1% (w/v) BSA for 1 h. After rinsing, the red fluorescence from tight junction proteins was observed by a Leica AF6000LX microscope equipped with a 63 × objective using ×1.6 magnifications.

**Statistical analysis**. All cell studies were done in triplicate incubations and each experiment was repeated at least 3 times. The results are presented as mean ± s.d. Statistical analysis and comparison of different groups in relation to one or two factors were performed with one-way ANOVA or two-way ANOVA as appropriate. The Bonferroni method was subsequently used to correct $p$ values after multiple comparisons to calculate statistical significance, otherwise stated.

**Complement activation**. NLC-mediated (final concentration of either 1 or 5 μM in serum) complement activation was performed in fresh human sera through ELISA determination of fluid phase C3a, C5a and sC5b-9 in human sera using respective Quidel kits (San Diego, CA, USA)[62]. Activation products did not adhere to FAM-CGY NLCs. Functional assessment of complement pathways were in accordance with manufacturer's specifications[62,63].

**Compliance**. Animal protocols were in accordance with the guidelines and regulations of "Animal Care and Use Committee Guidelines of Guangzhou Institute of Biomedicine and Health", approved and performed at Guangzhou Institute of Biomedicine and Health, China.

**Animal experiments**. Groups of male and females (50:50 distribution) ICR mice (6–8 weeks old, body weight 20–26 g) were randomly selected ($n = 3$ per group) and injected with Cy5.5-CGY NLCs, Cy5.5-CGY scramble peptide, Cy5-siRNA/FAM-CGY complexes, or Cy5-siRNA/FAM-CGY scramble peptide complexes via tail vein. The final concentration of the peptide conjugate was 10 μM and that of siRNA was 48 nM based on blood volume calculation equivalent to 6% of the body weight. At different time points blood samples were collected via retro-orbital blood collection. For whole animal imaging system mice were anaesthetised with isoflurane and in vivo fluorescence images were taken (IVIS 200 Spectrum, USA), at different time points. At 4 h post administration, mice were sacrificed, and the liver, lungs, kidneys, spleen, pancreas and brain were collected and stored in liquid nitrogen. The organs were homogenised (FastPrep-24™ 5G Instrument, MP Biomedicals, USA) and the levels of peptide and siRNA were measured in a fluorescence spectrometer (Fluotime 300, Picoquant, Berlin, Germany) using a standard curve.

In some experiments, animals were transcardially perfused with 10 mL 0.9% (w/v) saline at a rate of 2 mL min⁻¹ through the left ventricle to remove the blood from the brain, followed by 20 mL 4% (v/v) paraformaldehyde in phosphate buffered saline (PBS, pH = 7.4). The brain was removed and post-fixed overnight in 4% (v/v) paraformaldehyde followed by dehydration with 30% (w/v) sucrose solution in PBS at 4 °C. Frozen sections were cut on a cryostat (Thermo fisher, USA). The 10 μm thick brain sections were treated with 1% (w/v) BSA in PBS/0.1% (w/v) Tween 20 for 30 min, and stained with primary anti-RAGE (1:150 dilution), anti-Transferrin receptor (1:200 dilution), and anti-CD31 (1:400 dilution) antibodies individually overnight at 4 °C. Secondary antibodies were donkey anti-mouse Alexa Fluor®680 (1:1000 dilution) (in animals that received FAM-CGY NLCs; applicable to Fig. 4f) and donkey anti-mouse Alexa Fluor®405 (1:1000 dilution) (in animals that received Cy5-siRNA/FAM-CGY complexes; applicable to Fig. 4g) for microscopy detection. To visualise neurons, microglial and astrocytes, sections were first stained with antibodies against Tuj1 (1:500), Iba1 (1:300) and GFAP (1:300), respectively, and then with donkey anti-mouse Alexa Fluor®555 (1:1000 dilution) secondary antibody. DAPI staining was performed to visualise cell nuclei.

For in vivo BACE1 silencing, 20 ICR mice were randomly divided into five groups (4 mice in each group) and administrated intravenously with either 72 nM siControl/FAM-CGY or BACE1 siRNA (final concentration, 72 nM) or 72 nM siControl/FAM-CGY scramble peptide1 or 48 nM BACE1 siRNA/FAM-CGY or 72 nM BACE1 siRNA/FAM-CGY, respectively. The final concentration of peptide was 10 μM. After 48 h, mice were sacrificed and perfused with 10 mL 0.9% (w/v) saline at a rate of 2 mL min⁻¹ and brain was removed. The frozen brain tissues (80–90 mg) were homogenized in RIPA lysis buffer (Beyotime, China) and centrifuged at $18,000 \times g$, 4 °C for 15 min. Protein concentration of the supernatant was determined using the BCA Protein Assay Kit. Equal amounts of protein from each sample were loaded on SDS-PAGE gels (4–15%) for the determination with level of the target protein determined by Western blotting using the anti-BACE1 antibody (1:1000 dilution) and detected with HRP-Goat-anti-rabbit) IgG (H + L) secondary antibody (1:3000).

**Safety studies in animals**. The ICR mice were injected with FAM-CGY, BACE1 siRNA/FAM-CGY and Cy5-siRNA/FAM-CGY complex via tail vein with final peptide concentration of 10 μM and siRNA of 48 nM based on calculation of 6% of body weight for blood volume in mice. After 4, 24 and 72 h post injection, mice were sacrificed and perfused with 10 mL 0.9% (w/v) saline at a rate of 2 mL min⁻¹. Organs (brain, heart, liver, lung, kidney, spleen) were collected for haematoxylin-eosin (H&E) analysis. Tissues were sectioned at a 5 mm thickness.

H&E staining was done by submersion in Harris hematoxylin followed by differentiation through 37% HCl and $Li_2CO_3$ solution. An Olympus optical microscope (Tokyo, Japan) was used for assessing slides. In some experiments animals received an intravenous injection of 200 µg $kg^{-1}$ lipopolysaccharide and sacrificed at 24 h post injection for histological and blood count analysis.

Blood samples were collected for determination of red blood cell count, white blood cell count, haemoglobin, haematocrit, mean corpuscular volume, mean corpuscular haemoglobin, mean corpuscular haemoglobin concentration and platelet count (XFA6030, Perlong Medical Inc., China). Hippocampus was dissected out, rinsed in ice-cold PBS thoroughly, weighted and immediately frozen in liquid nitrogen. Tissues were cut into small pieces and homogenized in 1 mL ice-cold PBS with a glass homogeniser and sonicated with an ultrasonic cell disrupter to break the cell membranes. Homogenates were centrifuges for 5 min at $5,000 \times g$ and supernatants were assayed immediately for quantitative determination of TNF-α, IL-6, IL-1β, Iba-1 and GFAP by corresponding ELISA kits, respectively in accordance with manufacturer's instructions.

**Reporting summary**. Further information on research design is available in the Nature Research Reporting Summary linked to this article.

## Data availability

Raw data supporting peptide synthesis and characterisation, uncropped original SDS-PAGE gels (Fig. 1) and Western blots for BACE1 determinations in brain samples and TfR expression in human cells are reported in Supplementary Information file. Raw data for Figs. 2–6 is provided as the Source Data file. All other raw data supporting the findings of this study are available from the corresponding authors upon reasonable request.

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

## Acknowledgements
S.M.M. acknowledges financial support by the Danish Agency for Science, Technology and Innovation (Det Strategiske Forskningsråd) (reference 09-065746), Lundbeckfonden (reference R100-A9443) and International Science and Technology Cooperation of Guangdong Province (reference 2015A050502002) and Guangzhou City (reference 2016201604030050) with RiboBio Co, Ltd., China. L.-P.W. acknowledges financial support from Drug Discovery Pipeline of Guangzhou Institutes of Biomedicine and Health (reference 201508020131) and the National Science and Technology Major Projects for New Drug Development (reference 2018ZX09733-006).

## Author contributions
S.M.M. and L-P.W. conceived the idea and planned experiments. L.-P.W., D.A., J.S. and A.H. performed experiments. All authors designed, analysed and discussed data. S.M.M. wrote the paper with contributions from all co-authors.

## Competing interests
The authors declare the following competing interests: L.-P.W., D.A. and S.M.M. are named inventors on PCT, EP and US patent filings and US Patent 2019/0192657. Z.S.F. and S.M.M. declare financial interests in S. M. Discovery Group Inc. (USA) and S. M. Discovery Group Ltd. (UK). The remaining authors declare no competing interests.

## Additional information

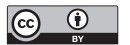

