## [Peer Review File · Nature Communications]

Reviewers' Comments:

Reviewer #1:

Remarks to the Author:

The manuscript "Self-assemblies from a phage display peptide as functional nanoligand drug carriers targeting two receptors and crossing the blood-brain-barrier" by Moghimi et al. describes the finding that a dodecapeptide selected by phage display requires self-assembly in order to bind the TfR target against which it was selected. Besides, the aggregated peptide, when forming fibres, binds another receptor (RAGE) that is known to take up aggregated proteins and that the combination of these two interactions provides specificity to brain endothelial cells. This finding would have great implications in the peptide display field as well as in the drug delivery field as it would explain why in many occasions peptides selected from phage display do not bind their target when grafted onto a nanocarrier in monomeric form.

However, I feel that the ground to substantiate this claim is at present premature and insufficiently supported by the data reported. I have listed the shortcomings below.

1. The authors have hypothesized that TfR targeting ability is conferred upon self-assembly of peptide amphiphiles, thereby exposing amino acid residues at a precise distance and orientation to enable TfR binding. To support this hypothesis the authors mention in line 180 that the amino acids R, Y, A, N and P that are known to be the main residues in Tf that mediate TfR binding are also present in the phage selected peptide. However, peptide self-assembly is driven by hydrophobic collapse, in which residues Y,A are most likely clustered and unavailable for receptor interaction. To verify the claim for self-assembling peptide-receptor interaction a more rigorous characterization is required. For example, by using ssNMR the supramolecular organization of the self-assembled peptide clusters could be elucidated, or molecular dynamics simulations could be applied to dock the peptide structures onto the TfR.

2. Moreover, to validate the claim of the conformational differences between the monomeric CGY and their respective assemblies, the authors are strongly advised to measure the CD spectrum of the monomeric (lower concentration, non-dye conjugated) as well as multimeric or aggregated peptides.

3. In the cell interaction studies section, to prove the peptide assemblies are specifically interacting with the TfR, the authors have performed competitive inhibition uptake studies with free Tf, uptake studies on hCMEC/D3 cells in which the TfR was silenced and differential uptake between MCF-7 and MCF-10A. We find a few shortcomings for these experiments to verify the authors claim---silencing of the TfR can also impair the endocytosis efficiency of the cell lines; the disparity of the endocytic efficiency between tumorigenic and non-tumorigenic cells can also lead to different uptake efficiency. Indeed, due to the structural similarity between A β oligomers and the FAM-CGY spherical aggregates, the authors are strongly advised to performed respective experiments to rule out the oligomer intake pathway. (Biochim Biophys Acta. 2008;1782(9):523-31).

4. The authors are strongly advised to perform the Tf receptor binding assay at 4 degrees Celsius as definitive proof for the specific assemblies-receptor interaction as well as the cell membrane interaction kinetics (which would be different between specific and unspecific interaction).

5. Knowing the A β oligomer are cytotoxic, the authors are strongly advised to perform cell viability assays (e.g. MTT/MTS) to test the cytotoxicity of FAM-GYR, FAM-CGY towards applied cell lines.

6. One of the conclusions drawn by the authors is: "We further demonstrate rapid.....and crossing of the blood-brain-barrier with NLCs, since both TfR and RAGE are highly expressed by cerebral endothelial cells and participate in ligand transcytosis." This claim is not substantiated enough by the results demonstrated. There seem to be colocalization with brain endothelial cells and to a certain extent with brain parenchymal cells, but this is not a definite guarantee that NLCs are being transcytosed and can reach the brain parenchymal cells intact. A functional assay to proof brain-specific delivery (e.g. BACE-1 silencing by delivering siRNA to neurons) will be needed to proof this.

Reviewer #2:

Remarks to the Author:

The manuscript authored by Wu and colleagues is interesting and innovative; however there are several limitations of the paper in the present form that should be addressed:

1) Internalization and functional studies were done in hCMEC/D3 cells in vitro. Additional studies are needed to evaluate the mechanisms involved in the passage of FAM-GYR complexes across the BBB. For that, authors should use an in vitro BBB model and evaluate tight junction integrity, permeability and further dissect mechanisms of endocytosis.

2) Authors do not show the effects of TfR and RAGE silencing in cell survival. These data should be presented.

3) RAGE receptors are upregulated in Alzheimer's disease, as stated in the manuscript, as well as in several inflammatory conditions. RAGE has been increasingly reported to trigger pro-inflammatory and cytotoxic effects and so involved in the pathogenesis of several diseases. Therefore, the in vivo paradigm should include experiments to disclose vascular permeability in selected brain regions, the analysis of inflammatory (microglia and astrocyte reactivity, the release of inflammatory mediators, etc) and cell death markers. Moreover, the authors should evaluate the internalization of FAM-GYR complexes by astrocytes, microglia and neurons. The peripheral toxicity should also be analyzed, for example, by using biomarkers of liver damage, kidney, brain, skeletal muscle and heart function.

4) The bioactivity (in terms of gene silencing) resultant from the delivery of siRNA by FAM-GYR was only demonstrated in vitro. A proof-of-principle should be also provided in in vivo.

Reviewers' comments:

Reviewer #1 (Remarks to the Author):

The manuscript "Self-assemblies from a phage display peptide as functional nanoligand drug carriers targeting two receptors and crossing the blood-brain-barrier" by Moghimi et al. describes the finding that a dodecapeptide selected by phage display requires self-assembly in order to bind the TfR target against which it was selected. Besides, the aggregated peptide, when forming fibres, binds another receptor (RAGE) that is known to take up aggregated proteins and that the combination of these two interactions provides specificity to brain endothelial cells. This finding would have great implications in the peptide display field as well as in the drug delivery field as it would explain why in many occasions peptides selected from phage display do not bind their target when grafted onto a nanocarrier in monomeric form.

However, I feel that the ground to substantiate this claim is at present premature and insufficiently supported by the data reported. I have listed the shortcomings below.

1. The authors have hypothesized that TfR targeting ability is conferred upon self-assembly of peptide amphiphiles, thereby exposing amino acid residues at a precise distance and orientation to enable TfR binding. To support this hypothesis the authors mention in line 180 that the amino acids R, Y, A, N and P that are known to be the main residues in Tf that mediate TfR binding are also present in the phage selected peptide. However, peptide self-assembly is driven by hydrophobic collapse, in which residues Y,A are most likely clustered and unavailable for receptor interaction. To verify the claim for self-assembling peptide-receptor interaction a more rigorous characterization is required. For example, by using ssNMR the supramolecular organization of the self-assembled peptide clusters could be elucidated, or molecular dynamics simulations could be applied to dock the peptide structures onto the TfR.

2. Moreover, to validate the claim of the conformational differences between the monomeric CGY and their respective assemblies, the authors are strongly advised to measure the CD spectrum of the monomeric (lower concentration, non-dye conjugated) as well as multimeric or aggregated peptides.

3. In the cell interaction studies section, to prove the peptide assemblies are specifically interacting with the TfR, the authors have performed competitive inhibition uptake studies with free Tf, uptake studies on hCMEC/D3 cells in which the TfR was silenced and differential uptake between MCF-7 and MCF-10A. We find a few shortcomings for these experiments to verify the authors claim---silencing of the TfR can also impair the endocytosis efficiency of the cell lines; the disparity of the endocytic efficiency between tumorigenic and non-tumorigenic cells can also lead to different uptake efficiency. Indeed, due to the structural similarity between A β oligomers and the FAM-CGY spherical aggregates, the authors are strongly advised to performed respective experiments to rule out the oligomer intake pathway. (Biochim Biophys Acta. 2008;1782(9):523-31).

4. The authors are strongly advised to perform the Tf receptor binding assay at 4 degrees Celsius as definitive proof for the specific assemblies-receptor interaction as well as the cell membrane interaction kinetics (which would be different between specific and unspecific interaction).

5. Knowing the A β oligomer are cytotoxic, the authors are strongly advised to perform cell viability assays (e.g. MTT/MTS) to test the cytotoxicity of FAM-GYR, FAM-CGY towards applied cell lines.

6. One of the conclusions drawn by the authors is: ‘We further demonstrate rapid.....and crossing of the blood-brain-barrier with NLCs, since both TfR and RAGE are highly expressed by cerebral endothelial cells and participate in ligand transcytosis.’ This claim is not substantiated enough by the results demonstrated. There seem to be colocalization with brain endothelial cells and to a certain extent with brain parenchymal cells, but this is not a definite guarantee that NLCs are being transcytosed and can reach the brain parenchymal cells intact. A functional assay to proof brain-specific delivery (e.g. BACE-1 silencing by delivering siRNA to neurons) will be needed to proof this.

Reviewer #2 (Remarks to the Author):

The manuscript authored by Wu and colleagues is interesting and innovative; however there are several limitations of the paper in the present form that should be addressed:

1) Internalization and functional studies were done in hCMEC/D3 cells in vitro. Additional studies are needed to evaluate the mechanisms involved in the passage of FAM-GYR complexes across the BBB. For that, authors should use an in vitro BBB model and evaluate tight junction integrity, permeability and further dissect mechanisms of endocytosis.

2) Authors do not show the effects of TfR and RAGE silencing in cell survival. These data should be presented.

3) RAGE receptors are upregulated in Alzheimer’s disease, as stated in the manuscript, as well as in several inflammatory conditions. RAGE has been increasingly reported to trigger pro-inflammatory and cytotoxic effects and so involved in the pathogenesis of several diseases. Therefore, the in vivo paradigm should include experiments to disclose vascular permeability in selected brain regions, the analysis of inflammatory (microglia and astrocyte reactivity, the release of inflammatory mediators, etc) and cell death markers. Moreover, the authors should evaluate the internalization of FAM-GYR complexes by astrocytes, microglia and neurons. The peripheral toxicity should also be analyzed, for example, by using biomarkers of liver damage, kidney, brain, skeletal muscle and heart function.

4) The bioactivity (in terms of gene silencing) resultant from the delivery of siRNA by FAM-GYR was only demonstrated in vitro. A proof-of-principle should be also provided in in vivo.

Reply to Reviewers' Comments

We are most grateful to both reviewers' for their constructive comments. We have carefully read the comments and rigorously addressed all raised issues and where necessary have performed new experiments and presented in the revised version.

Point-to-point reply:

Reviewer #1 (Remarks to the Author):

The manuscript "Self-assemblies from a phage display peptide as functional nanoligand drug carriers targeting two receptors and crossing the blood-brain-barrier" by Moghimi et al. describes the finding that a dodecapeptide selected by phage display requires self-assembly in order to bind the TfR target against which it was selected. Besides, the aggregated peptide, when forming fibres, binds another receptor (RAGE) that is known to take up aggregated proteins and that the combination of these two interactions provides specificity to brain endothelial cells. This finding would have great implications in the peptide display field as well as in the drug delivery field as it would explain why in many occasions peptides selected from phage display do not bind their target when grafted onto a nanocarrier in monomeric form.

Reply: We are grateful to the Reviewer for highlighting the importance of our work not only in relation to advanced drug delivery and brain targeting, but also in the phage display peptide field.

However, I feel that the ground to substantiate this claim is at present premature and insufficiently supported by the data reported. I have listed the shortcomings below.

1. The authors have hypothesized that TfR targeting ability is conferred upon self-assembly of peptide amphiphiles, thereby exposing amino acid residues at a precise distance and orientation to enable TfR binding. To support this hypothesis the authors mention in line 180 that the amino acids R, Y, A, N and P that are known to be the main residues in Tf that mediate TfR binding are also present in the phage selected peptide. However, peptide self-assembly is driven by hydrophobic collapse, in which residues Y,A are most likely clustered and unavailable for receptor interaction. To verify the claim for self-assembling peptide-receptor interaction a more rigorous characterization is required. For example, by using ssNMR the supramolecular organization of the self-assembled peptide clusters could be elucidated, or molecular dynamics simulations could be applied to dock the peptide structures onto the TfR.

Reply: Thank you. It is not the aim of this work to delineate the mode of interactions between the concerned receptors and the self-assemblies, but rather identification of the concerned receptors and whether self-assemblies can deliver nucleic acid cargos into cellular targets via these receptors at the BBB level.

The rationale for our previous statement was as follow:

The crystal structure of monoferric N-lobe human transferrin/TfR complex has suggested two possible binding motifs in the N lobe and one in the C lobe of human transferrin [Eckenroth, B. E., Steere, A. N., Chasteen, N. D., Everse, S. J. & Mason, A. B. How the binding of human transferrin primes the transferrin receptor potentiating iron release at endosomal pH. *Proc. Natl. Acad. Sci. USA* **108**, 13089–13094 (2011)]; although the exact nature of transferrin binding to TfR and subsequent dissociation steps still poorly understood. However, from the available information it appears that transferrin binding to TfR is through conformational motif, which may involve amino acids R, Y, A, N and P (Eckenroth et al. 2011). These amino acids are, indeed, present in the CGY peptide. In addition to CGY, a previously identified phage virion, which also targeted the TfR, contained the majority of these amino acids on its display peptide [Lee, J. H., Engler, J. A., Collawn, J. F., Moore, B. A. Receptor mediated uptake of peptides that bind the human transferrin receptor. *FEBS J.* **268**, 2004–2012 (2001)]. On the basis of these studies, we speculated on the role of the abovementioned amino acids in self-assembled nanoligand carrier (NLC) binding to the TfR; however, it was not our aim to identify the exact molecular steps in substrate-receptor binding and dissociation. In spherical NLCs, the protofilament projections are presumably arising from hydrophobic collapse and this potentially buries hydrophobic amino acids. However, we cannot disregard the presence of FAM-CGY monomers on the surface of core-shell NLCs, which may provide the additional epitopes (i.e., the aromatic amino acids) for binding to the TfR. Furthermore, by considering the fact that transferrin can efficiently compete with NLCs also indicate that NLCs have a lower affinity for the TfR than transferrin, despite being multivalent. In accordance with the aim of the present work, we have now removed the statement regarding involvement of the selected amino acids in TfR-NLC interaction.

However, we agree with the Reviewer that macromolecular and further biophysical studies could improve our understanding TfR-NLC interaction. Such investigations are beyond the scope of this paper and will have no bearing on the outcome of this work. Conceptually, we are dealing with development of a simple system that can cross the BBB and deliver nucleic acid medicines to the brain, where we have identified the involvement of at least two receptors in uptake and translocation processes. Considering the heterogeneous nature of the NLCs and the presence of variable proportions of the peptide monomer (in solution and in association with nanoparticle surfaces), core-shell nanoparticles and nanofibres as well as the stochastic surface properties of the spherical NLCs, it is very likely that ssNMR will provide broad non-discrete signals. Variable proportionality is also problematic for dynamic modelling (such as Coarse Grained simulations), where such simulations require introduction of numerous assumptions and permutations, which will further complicate conformational/binding assessment.

2. Moreover, to validate the claim of the conformational differences between the monomeric CGY and their respective assemblies, the authors are strongly advised to measure the CD spectrum of the monomeric (lower concentration, non-dye conjugated) as well as multimeric or aggregated peptides.

Reply: We have fully addressed this comment through addition of new CD spectral experiments, which substantiate our claim with respect to conformational differences between the monomeric CGY and FAM-CGY with self-assemblies. Under “Peptide synthesis and self-assembly” section we state:

“FAM-CGY exhibited a molecular mass of 2075.24 g/mol, a critical aggregation concentration (CAC) of 2.8 μ M in Milli-Q water (Supplementary Fig. 2) and a β -sheet structure at concentrations above CAC (Supplementary Fig. 3). At concentrations below CAC, FAM-CGY (and CGY) displayed a random coil conformation (Supplementary Fig. 3)”.

3. In the cell interaction studies section, to prove the peptide assemblies are specifically

interacting with the TfR, the authors have performed competitive inhibition uptake studies with free Tf, uptake studies on hCMEC/D3 cells in which the TfR was silenced and differential uptake between MCF-7 and MCF-10A. We find a few shortcomings for these experiments to verify the authors claim---silencing of the TfR can also impair the endocytosis efficiency of the cell lines; the disparity of the endocytic efficiency between tumorigenic and non-tumorigenic cells can also lead to different uptake efficiency. Indeed, due to the structural similarity between A β oligomers and the FAM-CGY spherical aggregates, the authors are strongly advised to performed respective experiments to rule out the oligomer intake pathway. (Biochim Biophys Acta. 2008;1782(9):523-31).

Reply: Thank you. We respectfully disagree with some of these comments, but have added further experiments to address internalisation routes of entry. First, the statement by the Reviewer that TfR silencing can impair endocytosis efficiency is not substantiated by references and is irrelevant to our aim. Here, we are assessing whether TfR plays any involvement in NLC uptake, where the results are compared with transferrin uptake (positive control) under identical conditions, and not a global assessment of endocytosis. However, the results still show considerable NLC uptake on TfR down-regulation, which may have occurred through other routes, thus indicating functional operation of other internalisation processes. To the best of our knowledge, the available published evidence [Wu, Y., Xu, J., Chen, J., Zou, M., Rusidanmu, A. & Yang, R. Blocking transferrin receptor inhibits the growth of lung adenocarcinoma cells *in vitro*. *Thoracic Cancer* **9**, 253–261 (2018)] suggest that TfR silencing could block the signal from the oncogene KRAS (a gene whose product K-RAS is involved in cell signalling pathways that control cell growth, cell maturation and cell death) in lung adenocarcinoma H1299 cells, which over-express TfR. Nevertheless, to substantiate our claim on the possible role for TfR in NLC uptake, competition studies with transferrin (Fig. 2c) have verified this and through subsequent experiments we identified RAGE as another possible receptor in NLC uptake.

Second, we are not concerned with global disparity in endocytic efficiency between tumourigenic (MCF-7) and non-tumourigenic (MCF-10A) cells, but rather exploiting the differences in TfR expression (one with high expression of TfR and the other with very low levels of TfR expression) among these cell lines to assess NLC uptake. Considering the dramatic differences in NLC uptake between MCF-7 and MCF-10A cells, we have now extended our studies with MCF-7 cells and show that saturation of TfR binding sites with holotransferrin at 4 °C dramatically diminishes NLC binding and clearly differentiates between NLC-specific binding to TfR and nonspecific interactions with plasma membrane (Fig. 2f). Thus, this experiment has confirmed that NLCs, at least, target TfR (and apparently this is a predominant NLC uptake route by MCF-7 cells).

We respectfully disagree with the Reviewer that there are structural similarities between A β oligomers and spherical NLCs. First, there is no sequence homology between the two species. Second, there are considerable morphological differences between A β oligomers and the spherical core-shell FAM-CGY NLCs. A β oligomers are globular structures of 5 nm in size [Dahlgren, K. N., Manelli, A. M., Steine Jr., W. B. Baker, L. K., Krafft, A. A. & LaDu, M. J. Oligomeric and fibrillar species of amyloid- β peptides differentially affect neuronal viability. *J. Biol. Chem.* **277**, 32046–32053 (2002)], whereas FAM-CGY NLCs are core-shell structures >100 nm, where the shell displays numerous protofilaments of variable lengths. The dimensional aspect also applies to self-assembled nanofibres (e.g., length \geq 200 nm). Size and shape are important determinant regulating nanoparticle internalisation routes, which may further explain involvement of multifaceted routes in NLC uptake. Nevertheless, we have now performed additional experiments, which further demonstrate involvement of multifaceted mechanisms in NLC transport processes. First, we show intracellular co-localisation of FAM-CGY NLCs with Texas Red-labelled transferrin (Supplementary Fig. 17). This provides further support for clathrin-mediated uptake processes. Second, studies in hCMEC/D3 cells with a panel of transport inhibitors show that the NLC uptake is

energy-dependent, confirms involvement of clathrin- and caveolae-dependent endocytic processes, and identify a role for macropinocytosis (Fig. 2b). The involvement of different NLC internalisation pathways is consistent with the polarised nature of endothelial cells and involvement of different receptors in NLC recognition. In contrast to FAM-CGY, the cellular uptake of FAM-GYR and other modified/scrambled FAM-conjugated peptides was relatively low (Supplementary Fig. 13), which further confirms that NLC binding to hCMEC/D3 cells is specific and structure-based.

The Reviewer may also note that in the previous submission we drove a predominant nanofibre assembly formation between FAM-CGY and Cy3-siRNA. These nanofibres provided a further opportunity to discriminate between RAGE- and TfR-dependent uptake mechanisms. To this end, first we showed poor uptake of free Cy3-labelled siRNA by hCMEC/D3 cells, whereas Cy3-siRNA was successfully taken up through nanofibre delivery (Fig. 2I). Competition studies demonstrated that the RAGE peptide ligand, but not transferrin, could comprehensively block FAM-CGY/Cy3-siRNA uptake (Fig. 3e), since the cellular levels of Cy3-siRNA were comparable to incubations challenged with free Cy3-siRNA. Therefore, RAGE appears to play an important role in the uptake of fibrous NLCs (200–500 nm ranges) at least in complexation with nucleic acids. This is in contrast to an earlier study [Chafekar, S. M., Baas, F. & Scheper, W. Oligomer-specific A β toxicity in cell models is mediated by selective uptake. *Biochim. Biophys. Acta-Mol. Basis Dis.* **1782**, 523–531 (2008)], showing HeLa cells and a human neuroblastoma cell line could not internalise fibrillar A β _{1–42}. These differences may be related to a considerably longer length of fibrillar A β _{1–42} (> 1 μ m) than NLCs (which rarely exceed 450 nm) and/or differences in receptor expression/functionality among different cells. Nevertheless, our observations may also explain why previously described amphiphilic peptide fibres had shown some specificity for the brain endothelium [Mazza, M. *et al.* Nanofiber-based delivery of therapeutic peptides to the brain. *ACS Nano* **7**, 1016–1026 (2013)] as the recognition process might have been mediated through RAGE binding due to its substrate specificity for large size multiple crossed β -sheet assemblies.

Finally, in accordance with suggestions of Reviewer 2, we further performed uptake and translocation studies using a validated hCMEC/D3 BBB model (please see our response to the question 1 by the Reviewer 2) to expand on translocation processes as well as confirming the integrity of tight junctions on NLC challenge.

In summary, our additional experiments together with previously receptor down-regulation (partial silencing without inducing cell death) and direct ligand competition studies have provided strong support for the involvement of TfR and RAGE in NLC recognition and uptake.

4. The authors are strongly advised to perform the Tf receptor binding assay at 4 degrees Celsius as definitive proof for the specific assemblies-receptor interaction as well as the cell membrane interaction kinetics (which would be different between specific and unspecific interaction).

Reply: Thank you. We have now performed these experiments. Thus, we have expanded our studies with MCF-7 cells, where the results in Fig. 2f show that saturation of TfR binding sites with holo transferrin at 4 °C dramatically diminishes NLC binding and clearly differentiates between NLC-specific binding to TfR and non-specific interactions with plasma membrane.

5. Knowing the A β oligomer are cytotoxic, the authors are strongly advised to perform cell viability assays (e.g. MTT/MTS) to test the cytotoxicity of FAM-GYR, FAM-CGY towards applied cell lines.

Reply: Thank you. The Reviewer may note that *in vitro* safety profiling was comprehensively addressed in the previous submission. Nevertheless, we have now expanded on this. First, MTT and related reduction assays are not necessarily reliable when compared with microscopic examination and ATP, DNA, or Trypan Blue determinations for various reasons as discussed previously [Huang, K. T., Chen, Y. H. & Walker, A. M. Inaccuracies in MTS assays: major distorting effects of medium, serum albumin, and fatty acids. *Biotechniques* **37**, 406–408 (2004); Wu, L-P., Ficker, M., Christensen, J. B., Trohopoulos, P. N. & Moghimi, S. M. Dendrimers in medicine: therapeutic concepts and pharmaceutical challenges. *Bioconjugate Chemistry* **26**, 1198–1211 (2015)]. Instead, we performed a comprehensive pan-integrated safety profiling based on plasma membrane integrity assessment and metabolic imprints. Thus, under “FAM-CGY complexation with nucleic acids” section we expand and state:

*“With respect to safety, neither FAM-CGY NLCs nor FAM-CGY/siRNA nanoparticles perturbed plasma membrane integrity as demonstrated through measurements of extracellular levels of lactate dehydrogenase (LDH) (Fig. 3c). In addition to this, we also employed an integrated metabolomics approach that measures ATP turnover and mitochondrial oxidative phosphorylation (OXPHOS) by high-resolution real-time respirometry. Calculations of ATP turnover, coupling efficiency of OXPHOS and respiratory control ratio (RCR) showed no significant changes on either FAM-CGY NLC or FAM-CGY/siRNA nanoparticle treatment compared with control (Fig. 3d, Supplementary Table 1). Collectively, these observations confer cell viability and safety, since OXPHOS accounts for the majority of ATP production in cells and mitochondrial dysfunction plays a major role in initiation of many types of cell-death processes. These results are in stark contrast to known cytotoxicity of A β oligomers [Chafekar, S. M., Baas, F. & Scheper, W. Oligomer-specific A β toxicity in cell models is mediated by selective uptake. *Biochim. Biophys. Acta-Mol. Basis Dis.* **1782**, 523–531 (2008)]. Finally, in contrast to FAM-CGY/siRNA nanocomplexes, siRNA delivery with siPORT was cytotoxic, it triggered LDH release and suppressed ATP turnover, OXPHOS and RCR (Fig. 3c & 3d, Supplementary Table 1)”.*

6. One of the conclusions drawn by the authors is: ‘We further demonstrate rapid.....and crossing of the blood-brain-barrier with NLCs, since both TfR and RAGE are highly expressed by cerebral endothelial cells and participate in ligand transcytosis.’ This claim is not substantiated enough by the results demonstrated. There seem to be colocalization with brain endothelial cells and to a certain extent with brain parenchymal cells, but this is not a definite guarantee that NLCs are being transcytosed and can reach the brain parenchymal cells intact. A functional assay to proof brain-specific delivery (e.g. BACE-1 silencing by delivering siRNA to neurons) will be needed to proof this.

Reply: Thank you. We have now addressed the *in vivo* bioactivity through BACE-1 siRNA delivery as suggested by the Reviewer (Section on “Brain targeting with intravenously injected NLCs”). Thus we measured the extent of β -secretase 1 (BACE 1) down regulation in hippocampus following intravenous injection of FAM-CGY/ β -secretase 1 (BACE 1)-specific siRNA complexes. BACE 1 is highly expressed in neurons and it is responsible for initiating A β generation and therefore has been considered as an important target for the therapeutic inhibition of A β production in Alzheimer’s disease. The results show that a single intravenous injection of FAM-CGY/BACE1-siRNA nanoparticles down regulates BACE 1 expression by ~50% compared with FAM-CGY/control siRNA complexes as determined by Western blot (Fig. 5c). Furthermore, intravenous injection of BACE 1 siRNA either in free form or with FAM-SP1 showed no pharmacological activity in hippocampus. These observations provide a pharmacological proof-of-concept for functionality of a nucleic acid medicine in the brain through NLC delivery and validate the capability of the engineered NLCs to function as neurological nanomedicines.

In the section “NLC Safety” we also used BACE-1 siRNA for forming spheroidal nanoparticles with FAM-CGY and assess in vivo safety and we show no inflammatory reactions and neuronal damage in the brain.

Reviewer #2 (Remarks to the Author):

The manuscript authored by Wu and colleagues is interesting and innovative; however there are several limitations of the paper in the present form that should be addressed:

1) Internalization and functional studies were done in hCMEC/D3 cells in vitro. Additional studies are needed to evaluate the mechanisms involved in the passage of FAM-GYR complexes across the BBB. For that, authors should use an in vitro BBB model and evaluate tight junction integrity, permeability and further dissect mechanisms of endocytosis.

Reply: We have performed additional experiments, which demonstrate multifaceted mechanisms in NLC transport processes (our response to point 3 by Reviewer 1). First, studies in hCMEC/D3 cells with a panel of transport inhibitors demonstrated that the NLC uptake is energy-dependent, and involves clathrin- and caveolae-dependent endocytic processes as well as macropinocytosis (Fig. 2b). The involvement of different NLC internalisation pathways is consistent with the polarised nature of endothelial cells and may suggest involvement of different receptors in NLC recognition. In contrast to FAM-CGY, the cellular uptake of FAM-GYR and other modified/scrambled FAM-conjugated peptides was relatively low (Supplementary Fig. 13). Collectively, the abovementioned observations further suggest that NLC binding to hCMEC/D3 cells is specific and predominantly structure-based.

Second, in line with the Reviewers' suggestion we further used a validated hCMEC/D3 BBB model. A hCMEC/D3 monolayer was formed and reached confluence on day 7 and the barrier integrity was confirmed through trans-endothelial electrical resistance (TEER) and cell layer capacitance (which reflects the membrane surface area) measurements with a CellZscope (Supplementary Fig. 19). TEER values were peaked on day 7 ($\sim 40\Omega\cdot\text{cm}^2$) in transwell inserts and longer culture of cells (up to 10 days) did not improve TEER and formation of multiple cell-layers was sometime observed from day 7 onward. Again, at day 7 the cell layer capacitance was below $2\ \mu\text{Fcm}^{-2}$ (compared with $\sim 10\ \mu\text{Fcm}^{-2}$ in empty inserts). Subsequently, we confirmed the expression of adheren β -catenin, which is required and has been reported for the formation of a functional tight junction (Supplementary Fig. 19). Thereafter, the apical side of the monolayers were challenged with NLCs, free 5-FAM and FAM-SP1 for 24 h. The data showed no detrimental effect of NLCs on TEER and cell layer capacitance and β -catenin distribution (Supplementary Fig. 19). This confirms preservation of the monolayer integrity following 24 h contact with NLCs. The data also shows NLC uptake by hCMEC/D3 monolayers as well as transport of a fraction across the monolayer (Supplementary Fig. 19). In contrast, neither the fluorophor 5-FAM nor the scrambled peptide showed notable transport across the BBB transwell (Supplementary Fig. 19).

In addition to the above, we further elaborated on the role of both TfR and RAGE in the uptake processes. Thus, we expanded our studies with MCF-7 cells as addressed by Reviewer 1. We performed a saturation study to further show that NLC uptake by MCF-7 cells is TfR specific. The results in Fig. 2f show that saturation of TfR binding sites with holo transferrin at 4 °C dramatically diminishes NLC binding and differentiates between NLC-specific binding to TfR and nonspecific

interactions with plasma membrane. The Reviewer may also note our comments to the Reviewer 1 (response to point 3) in relation to the predominant nanofibre assembly formation between FAM-CGY and siRNA and its mode of uptake, which has identified RAGE in the uptake of NLC nanofibres, at least in complexion with nucleic acids.

Based on the new experiments together with our previous dataset (competition studies and receptor down regulation attempts), we therefore conclude that NLC uptake by cerebral endothelial cells involves at least two receptors (TfR and RAGE) and consistent with both clathrin- and caveolae-dependent modes of internalisation. Macropinocytosis, which has been implicated in the uptake of protein aggregates by different cells, may still account for the uptake of various oligomeric/aggregate forms of FAM-CGY.

2) Authors do not show the effects of TfR and RAGE silencing in cell survival. These data should be presented.

Reply: Thank you. This was stated in the previous submission. Under “NCL-cell interaction” section we state that *“Using a commercial transfectant, TfR expression in hCMEC/D3 cells was down-regulated by 70% without inducing cell death (the cell viability was >95% as determined by the Trypan Blue exclusion test)”* and *“Also, lowering RAGE expression in hCMEC/D3 cells with a RAGE-specific siRNA (the cell viability was >95% on RAGE down regulation as determined by the Trypan Blue exclusion test) reduced NLC uptake compared with their respective mock-silenced cells (Fig. 2g).”*.

Furthermore, under “FAM-CGY Complexion with Nucleic Acids” section we performed a comprehensive safety profiling and state: *“With respect to safety, neither FAM-CGY NLCs nor FAM-CGY/siRNA nanoparticles perturbed plasma membrane integrity as demonstrated through measurements of extracellular levels of lactate dehydrogenase (LDH) (Fig. 3c). In addition to this, we also employed an integrated metabolomics approach that measures ATP turnover and mitochondrial oxidative phosphorylation (OXPHOS) by high-resolution real-time respirometry. Calculations of ATP turnover, coupling efficiency of OXPHOS and respiratory control ratio (RCR) showed no significant changes on either FAM-CGY NLC or FAM-CGY/siRNA nanoparticle treatment compared with control (Fig. 3d, Supplementary Table 1). Collectively, these observations confer cell viability and safety, since OXPHOS accounts for the majority of ATP production in cells and mitochondrial dysfunction plays a major role in initiation of many types of cell-death processes. These results are in stark contrast to known cytotoxicity of A β oligomers [Chafekar, S. M., Baas, F. & Scheper, W. Oligomer-specific A β toxicity in cell models is mediated by selective uptake. *Biochim. Biophys. Acta-Mol. Basis Dis.* **1782**, 523–531 (2008)]. Finally, in contrast to FAM-CGY/siRNA nanocomplexes, siRNA delivery with siPORT was cytotoxic, it triggered LDH release and suppressed ATP turnover, OXPHOS and RCR (Fig. 3c & 3d, Supplementary Table 1)”*.

3) RAGE receptors are upregulated in Alzheimer’s disease, as stated in the manuscript, as well as in several inflammatory conditions. RAGE has been increasingly reported to trigger pro-inflammatory and cytotoxic effects and so involved in the pathogenesis of several diseases. Therefore, the in vivo paradigm should include experiments to disclose vascular permeability in selected brain regions, the analysis of inflammatory (microglia and astrocyte reactivity, the release of inflammatory mediators, etc) and cell death markers. Moreover, the authors should evaluate the internalization of FAM-GYR complexes by astrocytes, microglia and neurons. The peripheral toxicity should also be analyzed, for example, by using biomarkers of liver damage, kidney, brain, skeletal muscle and heart

function.

Reply: We have addressed the Reviewers' suggestions with the following experiments:

- (a) Under "Brain targeting with Intravenously Injected NLCs" section we show *in vivo* association of both FAM-CGY (spheroidal nanoparticles) and FAM-CGY/Cy5-siRNA (nanofibres) NLCs with neurons and microglial cells, but not with astrocytes.
- (b) We have introduced a new section titled "NLC Safety". First, we studied morphological effects of intravenously injected FAM-CGY, FAM-CGY/siRNA and FAM-CGY/Cy5-siRNA NLCs in all main organs (Fig. 6). The haematoxylin and eosin images of brain sections (cerebral cortex and hippocampus) show that NLC treatment (regardless of NLC morphology and composition) does not induce neuronal injury/swelling, cellular liquefaction, necrosis and focal inflammatory cellular infiltration (Fig. 6a–d; Supplementary Fig. 23). In addition to these observations, ELISA assay analysis of brain homogenates for inflammatory markers tumour necrosis factor- α (TNF- α), interleukin-1 β (IL-1 β), IL-6, ionised calcium-binding adaptor molecule 1 (Iba-1; microglia marker) and glial fibrillary acid protein (GFAP; astrocyte marker) also revealed no inflammation in response to all NLC treatments (Fig. 6e–i).
- (c) In line with the brain investigations, no adverse morphological changes and inflammatory reactions were observable in the liver and the kidneys (Supplementary Fig. 24) as well as other major organs (Supplementary Fig. 25) at all studied time points and the results were similar to those of control animals. These observations are particularly important with respect to the liver and the kidney, as these organs play a key role in the overall NLC extraction from the blood. Furthermore, NLC treatment had no effect on blood count (Supplementary Table 2). As a positive control we show organ damage and alterations in blood count on lipopolysaccharide injection. Finally, NLCs did not induce complement activation in human sera (Supplementary Fig. 26). Collectively, these observations together with lack of *in vitro* cytotoxicity and metabolomics interference of FAM-CGY in hCMEC/D3 cells confer NLC safety.
- (d) With respect to the above, the Methods section has been expanded providing detailed experimental protocols for all tests.

4) The bioactivity (in terms of gene silencing) resultant from the delivery of siRNA by FAM-GYR was only demonstrated *in vitro*. A proof-of-principle should be also provided *in vivo*.

Reply: Thank you. The Reviewer 1 has also raised this point (please see our response to point 6 by Reviewer 1). Briefly, we show *in vivo* bioactivity by measuring the extent of β -secretase 1 (BACE 1) down regulation in hippocampus following intravenous injection of FAM-CGY/ β -secretase 1 (BACE 1)-specific siRNA complexes. The results show that a single intravenous injection of FAM-CGY/BACE1-siRNA nanoparticles down regulates BACE 1 expression by ~50% compared with FAM-CGY/control siRNA complexes as determined by Western blot (Fig. 5c). Furthermore, intravenous injection of BACE 1 siRNA either in free form or with FAM-SP1 (scrambled peptide) showed no pharmacological activity in hippocampus.

Reviewers' Comments on Revision:

Reviewer #1 provided confidential comments to the editor supportive of publication.

Reviewer #2 (Remarks to the Author):

The authors had adequately addressed all the issues raised. Therefore, I recommend the acceptance of the manuscript in the present form.

Reviewers' Comments:

Reviewer #1:

None

Reviewer #2:

Remarks to the Author:

The authors had adequately addressed all the issues raised. Therefore, I recommend the acceptance of the manuscript in the present form.